# Automatic optimization of flat-field corrections by evaluation and enhancement (EVEN) in multimodal optical microscopy

Elena Corbetta [1,2,3], Matteo Calvarese[1], Patrick Then[1,4], Hyeonsoo Bae [1], Tobias Meyer-Zedler [1], Bernhard Messerschmidt[5], Orlando Guntinas-Lichius [6], Michael Schmitt [2,3], Christian Eggeling [1,4], Juergen Popp[1,2,3] & Thomas Bocklitz [1,2,3] ✉

Uneven illumination affects all images acquired by optical microscopes, especially large, multicolour and nonlinear measurements. Although removal is possible with various algorithms, evaluating raw and processed images is challenging due to the lack of established workflows for image quality assessment. This manuscript describes a machine learning-based method, EVEN (Evaluation and Enhancement), to assess and optimise corrections in optical microscopy. EVEN integrates quantitative image metrics into a Linear Discriminant Analysis model to detect and predict image quality, automatically optimising corrections. The method can be integrated into the optical microscopy pipeline to simplify further processing and analysis. Here, we show the implementation and application of EVEN in different processing scenarios, including multimodal nonlinear imaging of human and neck tissue slices and multichannel fluorescence measurements of stained cells, demonstrating its capability to automatically optimise image quality by assessing single-channel corrections.

Optical microscopy plays a crucial role in biological and biomedical research, from the investigation of fundamental biological processes to the integration of microscopy techniques in clinical scenarios[1–4]. In these applications, the presence of experimental artifacts significantly impacts the quality of the measurement and the extraction of quantitative information for a correct understanding of the sample. Among the possible causes of image degradation, such as noise, blurring, and photobleaching, uneven illumination affects all images acquired by optical microscopes. It is a non-uniform illumination of the field of view (FOV), typically arising as a decrease of intensity from the optical axis to the edges of the measured image, resulting in a vignetting effect in single fields of view. In addition, when neighbouring tiles of a large

sample are stitched, the resulting composite image is affected by a mosaic effect. Uneven illumination not only generates strong artifacts that impede the correct visual interpretation of the images, but it might limit the performance of image analysis methods[5–8].

Uneven illumination is particularly strong in certain experimental conditions. For instance, nonlinear processes transmit any non-uniformity of the illumination signal to the generated and detected signal in a nonlinear way, whereas multimodal and multicolour measurements might show a variable illumination in the different channels due to the occurrence of different physical processes, chromatic aberrations, and the presence of different targeted tissue components. Multimodal measurements encode important morpho-chemical

[1]Leibniz Institute of Photonic Technology, Member of Leibniz Health Technologies, Member of the Leibniz Centre for Photonics in Infection Research (LPI), Jena, Germany. [2]Institute of Physical Chemistry (IPC), Friedrich Schiller University Jena, Member of the Leibniz Centre for Photonics in Infection Research (LPI), Jena, Germany. [3]Abbe Center of Photonics (ACP), Friedrich Schiller University Jena, Member of the Leibniz Centre for Photonics in Infection Research (LPI), Jena, Germany. [4]Institute of Applied Optics and Biophysics, Friedrich Schiller University, Jena, Germany. [5]GRINTECH GmbH, Jena, Germany. [6]Department of Otorhinolaryngology, Jena University Hospital, Jena, Germany. ✉e-mail: thomas.bocklitz@uni-jena.de

information, that is degraded and lost if the images are analysed without proper correction of uneven illumination artifacts[9]. In this regard, the removal of uneven illumination becomes essential for spectral histopathology studies. Fig. 1a shows a standard experimental pipeline for microscopic images affected by uneven illumination. The pipeline is implemented to retrieve quantitative information from the measured sample sections, including reconstruction steps for the generation of the raw image and processing algorithms to remove the artifacts before the final image analysis.

Regarding the image processing step, state-of-the-art algorithms have been implemented for flat-field correction and effective reduction of uneven illumination. These methods can be divided into spatial domain methods, frequency domain methods, stitching approaches, and artificial neural network-based corrections. Spatial domain methods estimate the illumination function and then correct the artifacts introduced by the optical system by inverse modelling. Within this category, there are multiple approaches to determine the illumination function. An estimated illumination mask can be obtained, for example, through an initial calibration by measuring a known sample with the same experimental conditions used for the experiment. Alternatively, the illumination mask can be computed from the measured images by processing the single tiles (prospective methods) or by implementing iterative approaches to retrieve the underlying illumination function (retrospective methods)[5–8,10,11]. Frequency domain methods are applied in the Fourier space, by locating and reducing the effects generated by the mosaic artifact in the frequency domain[9,12]. Stitching methods are algorithms that remove the mosaic artifacts in composite images by improving the stitching process and enhancing the intensity similarity between neighbouring regions. They can be paired with other correction methods for an improved result[9,13]. Lastly, the use of artificial neural networks has gained interest to tackle not only the correction of periodic artifacts in stitched images, but also

non regular stripes, out-of-focus regions, and bubbles[14,15]. The correction of uneven illumination can be also executed by non-experience users thanks to user-friendly tools, as many approaches are implemented as Fiji and Napari plugins[5,10,16,17].

Despite the availability of these methods, their parameters must be properly tuned. Conventional approaches often involve a long, time-consuming series of trials and errors, and incorrect parameter selection can result in suboptimal correction. In addition, the consequences of uneven illumination are often underestimated, and the assessment of the corrected images remains an open issue. To the best of our knowledge, there are nowadays no established, automated pipelines for the quantification of uneven illumination and for an objective evaluation of raw and processed images. Quality assessment is challenging due to the unavailability of the ground truth, despite the possibility in some cases to run initial calibrations. This issue is underlined by previous studies, where the comparison between different correction methods was assessed visually or by using overlapping images[5,9]. Visual assessment from an expert evaluator can be valid, but not objective and quantitative, while the latter approach is time consuming and requires significant overlap between tiles; therefore, it is a very effective approach for methods validation, but it is not practical in real-case experimental scenarios due to its complex implementation and the need of evaluating strong asymmetries in the FOV that extend further than the overlap region, especially for point-scanning systems and nonlinear techniques.

In this paper, we present a machine learning (ML)-based automated workflow for quantitatively assessing uneven illumination in raw and corrected images. The method, named EVEN, is intended to Evaluate the images of interest, and use the result to Enhance their quality by determining which is the optimal result among the generated corrections. Therefore, EVEN is a quality evaluation and optimization tool that makes the intensity distribution uniform from both a

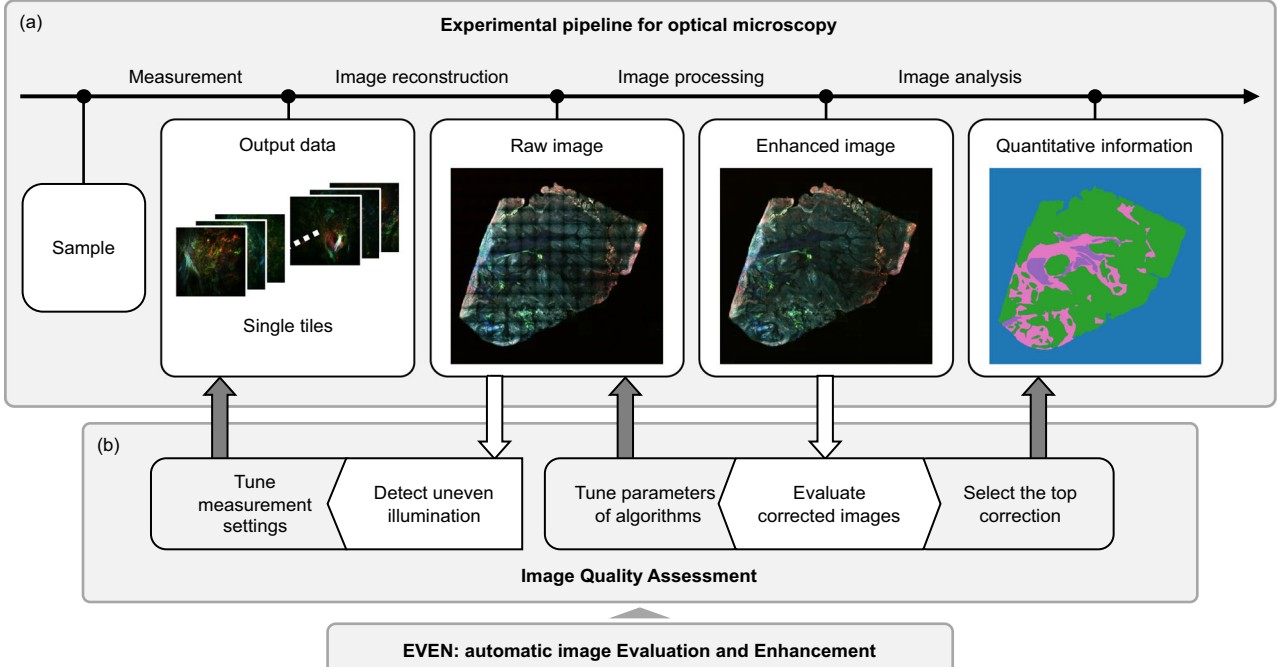

**Fig. 1 | Experimental pipeline for optical imaging of large samples in presence of uneven illumination. a** The experimental pipeline for optical microscopy is composed of image reconstruction, processing, and analysis steps to extract quantitative information from the measured sample and address the questions raised by the research study. When uneven illumination affects the measured images, the pipeline must be adapted to remove this artifact and enable an accurate image analysis. In the case of measurements of large samples, multiple tiles of the tissue are stitched into a composite image and flat-field correction is applied. Finally, the enhanced image can be analysed to retrieve the information of interest. **b** Image quality assessment should be always integrated in the experimental pipeline for a quantitative evaluation of the raw and corrected images. It can be utilized for automatic detection of uneven illumination, fine-tuning of the hyperparameters of different correction methods and the selection of the optimized correction before image analysis. EVEN can be used as a reliable and automatic tool for these tasks.

visual and quantitative, objective perspective. EVEN can be trained on a small dataset and generalised over different samples and imaging modalities. It can be easily integrated in any experimental pipeline for optical microscopy to address different evaluation issues (Fig. 1b), as we demonstrate in this manuscript: EVEN can be utilized to detect vignetting and adjust the acquisition parameters during the measurement process, fine-tune hyperparameters of single correction methods, compare different processing methods, and select the best output before image analysis. The application of EVEN is particularly advantageous for multi-modal and multichannel images, where each channel might require an individual optimization. In this manuscript, we demonstrate EVEN's effectiveness and generalizability by applying it to images processed by established correction methods and including datasets from varied sources.

## Results

This study proposes a quantitative image evaluation workflow that can be utilized to assess and improve the correction of uneven illumination in experimental images. In this section we present the steps for the implementation of EVEN, as summarized in Fig. 2, we validate the method by optimizing the correction of uneven illumination in non-linear multimodal measurements of human head and neck tissue, and we provide further application results to demonstrate the generalizability of EVEN.

### Step 1: definition of the evaluation criteria

Having a general understanding of the effects of uneven illumination on measured images, we can define specific quality metrics to mark the presence of vignetting and mosaic effect (Fig. 2a) and assess their correlation with the presence of the artifact using simulated images and a known experimental dataset from a previous study[9].

As shown in Fig. 3, two metrics are selected for the detection of uneven illumination in composite images made of multiple tiles. The edge energy ratio $E_{edge}$ (Fig. 3a) evaluates the presence of vignetting in single tiles by estimating the fraction of image energy measured at the borders of the FOV. For an image composed of multiple tiles, the overall $E_{edge}$ is the ratio computed for the sum of the tiles. The positive prominence $P_+$ (Fig. 3d) estimates the strength of the mosaic effect in the composite image as the sum of prominence of single periodic peaks that are generated in the power spectrum by the periodic grid in the direct domain.

The plots in Fig. 3 demonstrate that the metrics change significantly in presence of uneven illumination. Fig. 3b, e show the two metrics computed for 10 experimental high-quality images with 5 increasing levels of simulated uneven illumination. Fig. 3c, f show the statistical difference of the metrics computed for a known experimental dataset composed of 40 measurements affected by strong uneven illumination and 40 high-quality images that were well corrected by different processing methods (see subset of the images in Supplementary Fig. 1 and details about the correction in the Methods). A good removal of uneven illumination is correlated with high $E_{edge}$ and low $P_+$.

### Step 2: automatization and interpretation of the evaluation

Once we have assessed the correlation between the metrics and uneven illumination, we utilize the metrics as features to train a binary classification model to differentiate between images with and without uneven illumination. The trained model will be used not only to detect the presence of uneven illumination in raw images and determining the successful removal of the artifacts after processing, but more importantly to assign a score to every correction and automatically rank the images. To this aim, we train a Linear Discriminant Analysis (LDA) model on a set of experimental single-channel nonlinear measurements, consisting of 40 good and 40 bad images composed of multiple tiles (workflow of Fig. 2b and Supplementary Fig. 1). The bad

images are experimental measurements affected by strong uneven illumination, and the 40 good images are well corrected images without additional artifacts generated during the correction process. LDA is a simple and efficient approach known for low risk of over-fitting, ideal to obtain a generalizable model when the training dataset is limited. Moreover, LDA classifies the images by generating an interpretable decision score, which is the linear combination of the input features weighted by class coefficients and means (Supplementary Fig. 2). The score can be used for the generation of quality rankings (see the Methods section). The model achieves an average accuracy of 0.81, sensitivity (true rate for good images) of 0.95, and specificity (true rate for bad images) of 0.67.

The trained model is utilized to predict presence of uneven illumination in an unseen experimental dataset and automatically select the good corrections without prior visual assessment. The prediction dataset is composed of 23 three-channel measurements of human head and neck tissue slices, measured with coherent anti-Stokes Raman scattering (CARS), second harmonic generation (SHG) and two-photon excitation fluorescence (TPEF)[18]. Raw images are corrected by three established methods in the direct and Fourier domain (BaSiC[10], CIDRE[5] and Fourier method[9,12]) and predicted by the model. To assess agreement with visual perception, images are labelled according to their visual quality. The model scores a balanced accuracy of 0.74, sensitivity of 0.84 and specificity of 0.62 compared to the visual assessment. Compared to other ML-based classifiers, LDA shows good performance for both the 5-fold cross-validation with the training dataset and the prediction of unseen images, as well as good agreement with alternative linear classifiers, which show good generalization for our problem (Supplementary Fig. 3). The model shows an acceptable stability when applied to a different set of measurements, with a small bias towards good images. It's worth noting that visual evaluation might be based on additional features of the images, like the appearance of shading artifacts or the background contribution, while the model can evaluate only the presence of vignetting and the mosaic effect. Moreover, the prediction dataset does not always show a clear split between good and bad images as the training dataset. This underlines the importance of the generation of a quality ranking for each image to enable a more comprehensive evaluation.

To implement an in-depth evaluation workflow, we interpret the decision score assigned by the trained model to each image as a quality score. In a binary classification approach, the good class is associated to positive decision scores, while bad images obtain negative scores. The comparison of relative scores of different single-channel corrections enables the generation of automatic quality rankings and the investigation of specific classification trends, shown in Supplementary Fig. 4. Raw images are clearly characterized by lower scores, mostly with negative value. CIDRE obtains a positive score for TPEF channel, and the Fourier method obtains a positive score for most CARS and TPEF images. Instead, BaSiC shows a greater variability, with diversified values across different samples and channels. Despite the lower variability of CIDRE and Fourier for specific channels, the method is not biased against any correction method and shows mutable rankings for single-channel images. We provide in Supplementary Fig. 5 the values of the quality metrics for the prediction dataset for further interpretation of specific cases.

### Step 3: automatic optimization of uneven illumination corrections

Prediction results demonstrate that the metrics can be successfully implemented to detect the presence of uneven illumination, and that the trained model provides not only the classification label but also a quality score for each input image. In this paragraph, we further explore the ML-based approach to generate an automatic quality evaluation and optimization of corrected multimodal measurements.

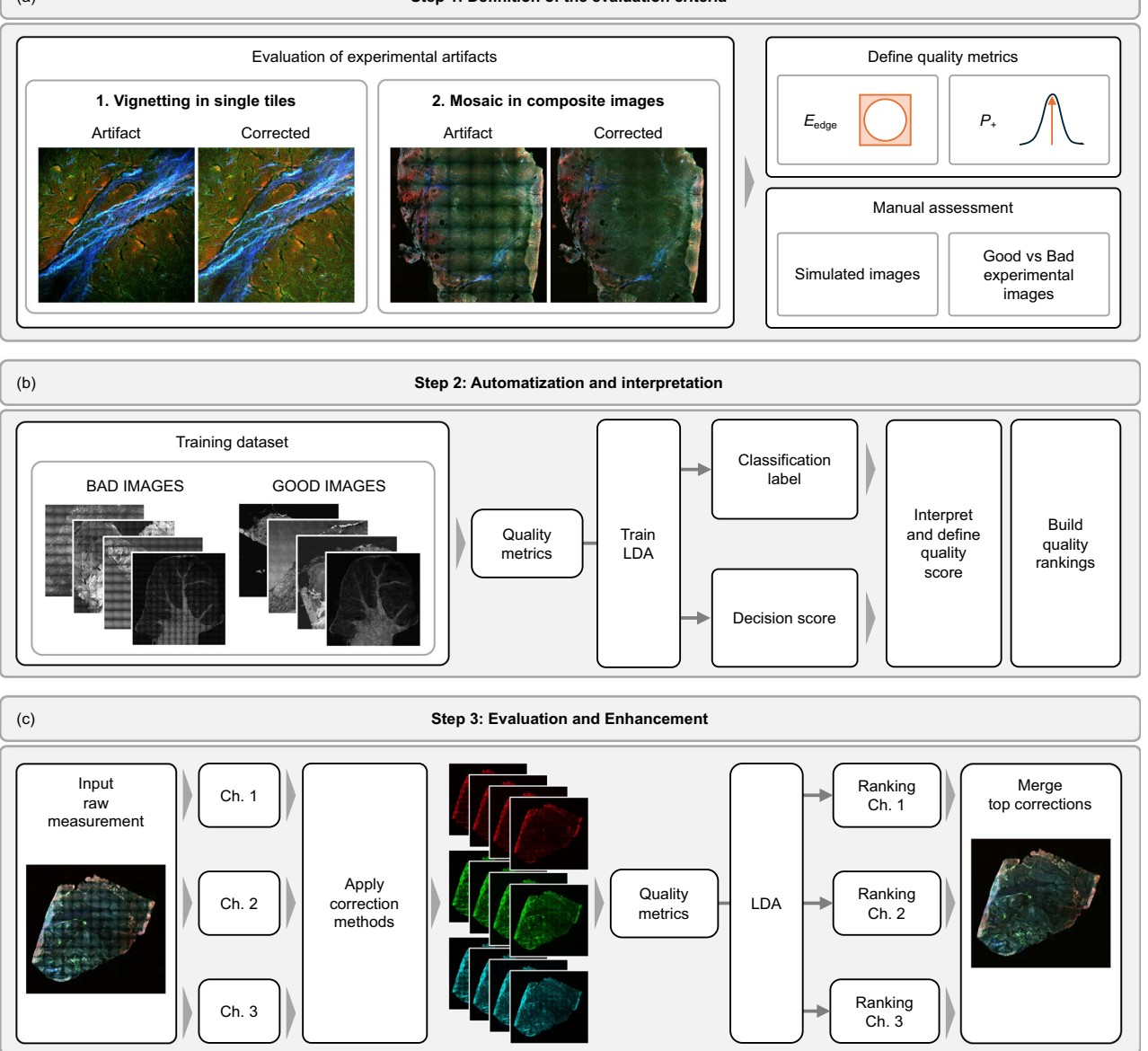

**Fig. 2 | Making uneven illumination EVEN: from the preliminary Evaluation of experimental artifacts to the automatic Enhancement of experimental images. a** Step 1: Definition of the evaluation criteria. We select quantitative metrics to detect vignetting in single fields of view and the mosaic effect in large images composed by multiple tiles: edge energy ratio $E_{edge}$ and positive prominence $P_+$. The metrics are assessed manually on semi-synthetic images and on experimental measurements. **b** Step 2: Automatization and interpretation. The identification of uneven illumination is automatized by training a Linear Discriminant Analysis (LDA) model to differentiate images with and without uneven illumination. The quality metrics are used as features to train the model. The coefficients of the trained model can be analysed to investigate how the metrics are utilized to determine the presence of uneven illumination, while the decision score can be interpreted as a quality score to build a quality ranking of the input images. **c** Step 3: Evaluation and Enhancement of unseen images. Multimodal images are automatically optimized by exploiting the LDA prediction. The single channels (Ch. 1, 2, 3) of the raw input image are corrected independently with multiple correction methods, then they are provided as input to predict image rankings for the raw and corrected versions of each channel. The top image for each channel is selected automatically to generate an optimized multimodal image.

Multimodal images provide morpho-chemical information about the different tissue components, that generate a distinct intensity distribution across the different channels (Fig. 4). In addition, the signals are generated by distinct physical processes, thus inducing even more variable uneven illumination effects. For these reasons, the best correction might be achieved using different correction methods or by fine-tuning the hyperparameters of the same algorithm for each channel. EVEN provides a flexible approach for the optimization of multimodal images, as shown by the workflow in Fig. 2c. In the full optimization pipeline, the input image is a raw measurement affected by uneven illumination, whose channels are independently corrected by multiple methods. Then, the corrected single-channel images are predicted by the trained LDA model, and a quality ranking is generated for each channel. Finally, the leading images of single-channel rankings are selected for the generation of the optimized multimodal image.

Figure 4 shows the application of EVEN to a single measurement of the dataset. The input image (Fig. 4a) is corrected by BaSiC, CIDRE and the Fourier method, that are applied in parallel to the single channels (Fig. 4b). In the lower part of Fig. 4b we observe that the corrections obtained by single methods might leave residual uneven illumination in at least one channel, because each method performs differently on the single intensity distributions: for example, CIDRE

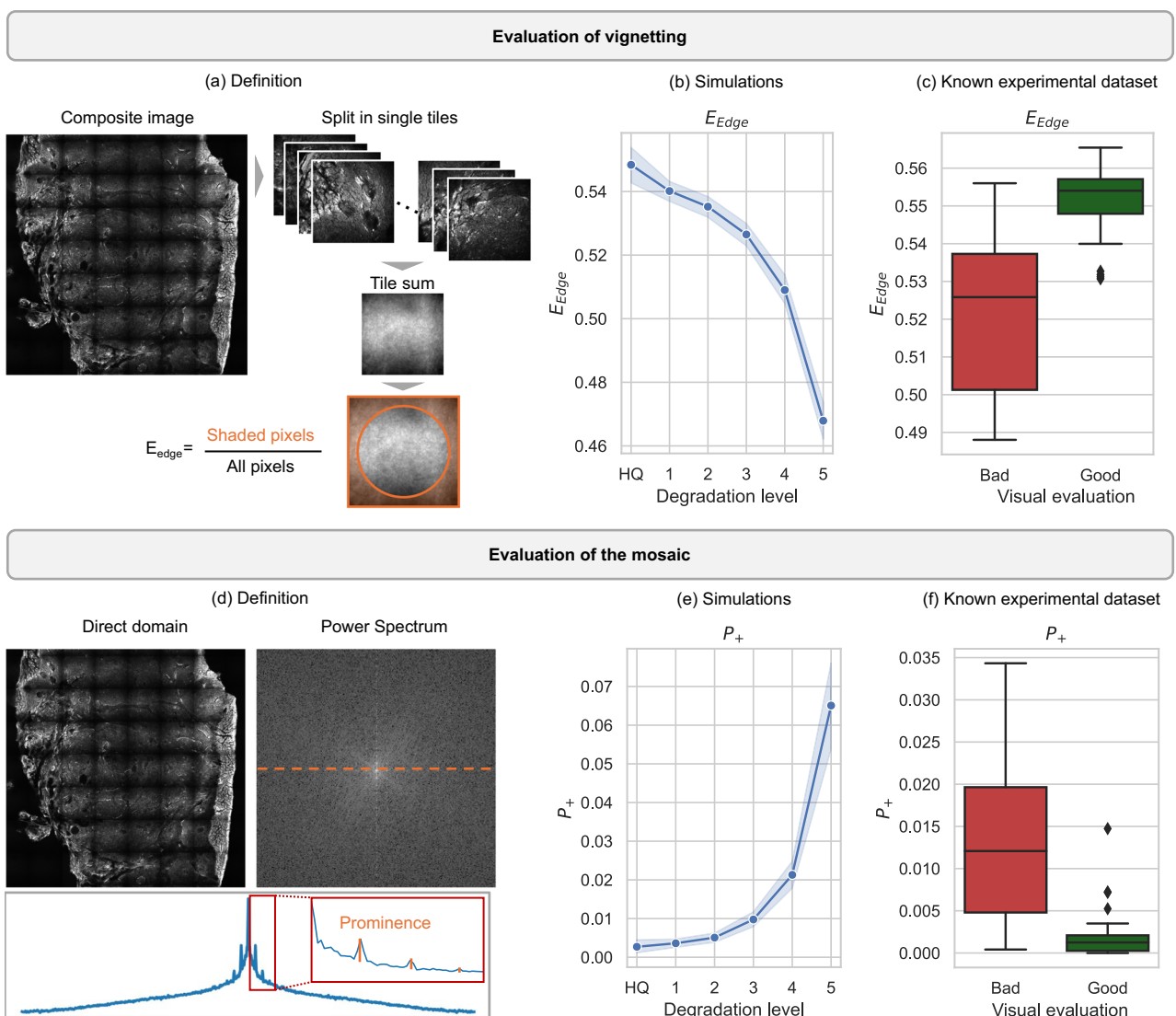

**Fig. 3 | Manual assessment of uneven illumination by quantitative metrics.**
**a** Workflow for calculating the edge energy ratio ($E_{edge}$) to detect vignetting in composite images: the image is split into single tiles, that are summed upon computation of the ratio between the sum of selected shaded pixels (marked in orange) and all pixels. **b** Trend of $E_{edge}$ for semi-synthetic images with 5 increasing levels of uneven illumination. HQ is the high-quality image without any artifact. **c** Value of $E_{edge}$ for known experimental images (training dataset) with (bad images, red) and without (good images, green) uneven illumination. A low value is correlated with the presence of strong artifacts. **d** Representation of the periodic peaks generated in the frequency domain by the mosaic artifact in composite images. The profile (orange dashed line) extracted from one of the main axes of the power spectrum, plotted in logarithmic scale, shows the positive prominence ($P_+$) of the peaks over the baseline of the power spectrum (orange line in the red squared box, that shows a zoomed area). **e** Trend of $P_+$ for 10 semi-synthetic images with 5 increasing levels of uneven illumination. HQ is the high-quality image without any artifact. **f** Value of $P_+$ for known experimental images (training dataset) with (bad images, red) and without (good images, green) uneven illumination. High prominence is correlated with the presence of strong artifacts. Shaded error bars in (b,e) show the 95% confidence interval around the mean value, marked by the central line, for metrics computed for $n = 10$ images with same artifact level. Boxplots in (**c**, **f**) indicate median (middle line), first and third quartile (box); whiskers extend until the 1.5 interquartile range and outliers are represented as single points. Minimum and maximum values are, respectively, 0.49 and 0.57 for $E_{edge}$, and 2.12e-7 and 3.34e-2 for $P_+$.

leaves strong residual artifacts in the CARS channel but it shows the best performance for SHG. Instead, the optimized correction predicted by EVEN shows a better quality and no residual artifacts (Fig. 4c), resulting in the Fourier correction for CARS, and CIDRE correction for TPEF and SHG, with good agreement with visual perception. We would like to emphasize that this prediction is not a general evaluation of the performance of the algorithms, but rather an evaluation of their application to a specific multimodal image with selected hyperparameters. In Supplementary Fig. 6–28 we compare the optimized results for the whole prediction dataset with the raw images and single-method corrections, demonstrating that EVEN is reliable and flexible in the selection of the top correction. EVEN shows

a good optimization performance for most samples, being able to generate multimodal images with a higher quality compared to the single-method corrections. When residual uneven illumination is present in all available corrections, the EVEN output is the best compromise among the available combinations (Supplementary Figs. 13, 14, 17). Indeed, EVEN performs a good optimization within the limits set by the correction methods and cannot compensate for residual artifacts left by all methods. In this cases, further processing would be necessary to optimize visual interpretation and automatic analysis, but for the demonstration of EVEN working principle it is beneficial to show the result also in presence of non-optimal images. In few cases (Supplementary Figs. 8, 12, 22, 27) all corrections are visually equivalent and

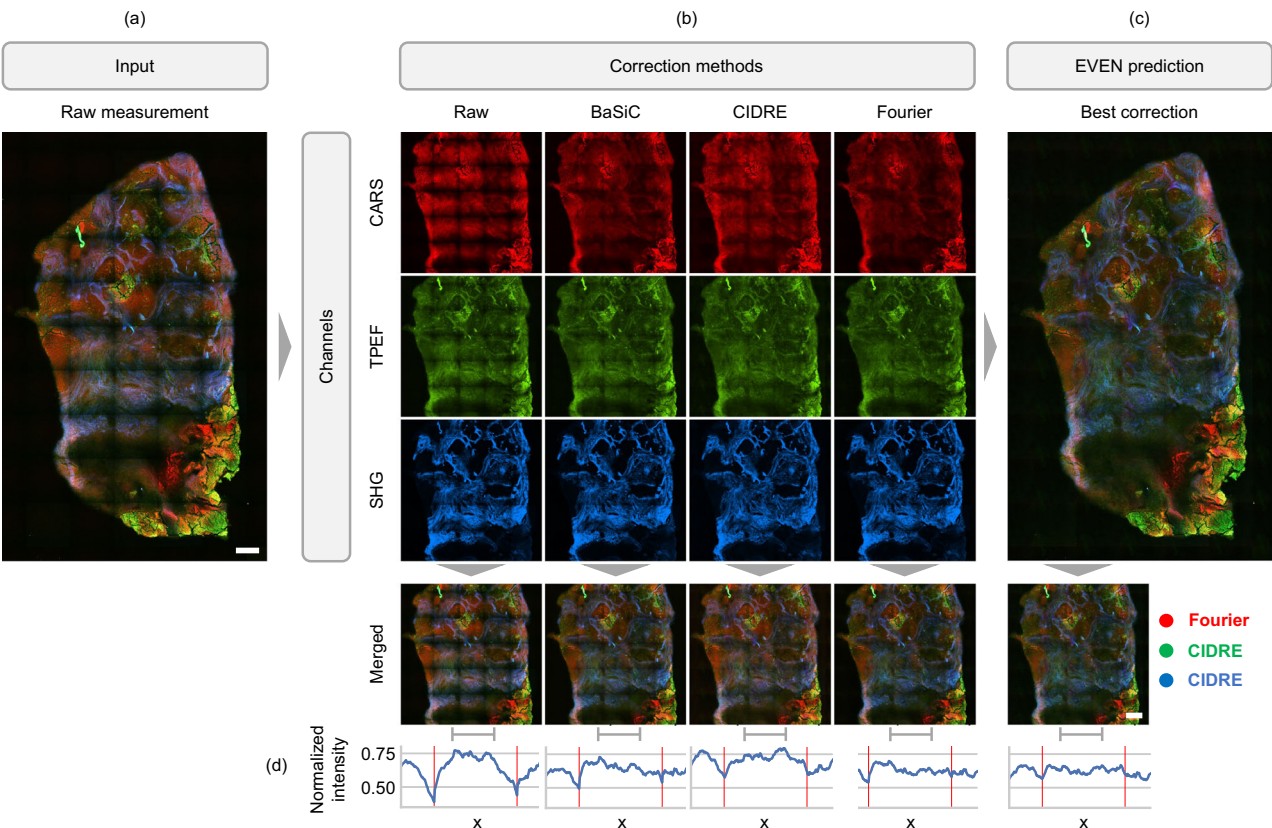

**Fig. 4 | Evaluation and enhancement of multimodal nonlinear measurements of human head and neck tissue.** One of the multimodal images of prediction dataset 1 is automatically optimized by EVEN. **a** The raw input measurement with strong uneven illumination. **b** The three channels (red: CARS; green: TPEF; blue: SHG), are corrected independently with three different correction methods: BaSiC[10], CIDRE[5], Fourier[9]. Due to different intensity distribution in the channels, that are generated by different morphological features of the different components of the tissue, the correction methods show a different performance for different channels. Indeed, the multimodal corrections obtained by single methods (bottom row) are different and show residual artifacts. **c** The optimized correction is predicted by EVEN after the computation of the quality metrics for each correction of the channels. For this specific measurement, the Fourier method is selected as the best correction for the CARS signal, as it provides flatter intensity distribution and enhancement of few bright regions without loss of details in the darker areas, whereas CIDRE is selected for TPEF and SHG channels. The optimized image shows a stronger reduction of uneven illumination compared to the single-method cases. **d** Intensity profile along x in the section of the image marked by the grey ruler. For each image, we generated a grayscale sum of the three channels and then plotted the maximum normalized intensity of the pixel sum along y. The red vertical lines mark the border of the tiles. EVEN shows the strongest reduction of vignetting at the tiles borders and the flattest profile. Scale bar: 200 μm.

indeed most of their channels show a low variability in the decision scores of Supplementary Fig. 4 and quality metrics of Supplementary Fig. 5, demonstrating that EVEN do not generate unexpected results for high-quality images. Lastly, our method fails just in the optimization of one sample (Supplementary Fig. 28), whose BaSiC correction appears better than EVEN output; however, neither correction can eliminate non-uniform illumination, and all images are characterised by a high decision score, probably due to a high edge energy generated by the artifacts (Supplementary Fig. 4). Therefore, a fine-tuned optimization of the set of quality metrics might be required in case of unusual behaviour of the correction methods.

We provide additional examples of experimental images optimized by EVEN, to show that our method is generalizable to other applications. Fig. 5 shows the optimization of a three-channel measurement of stained HEK293 cells. EVEN selects the CIDRE correction for the red and green channel, and the Fourier correction for the blue channel. The generated images are further analysed by Cellpose[19] to obtain automatic cell segmentation and assess the improvements provided by EVEN (Fig. 5b). Figure 5c shows intensity profiles generated by summing the channels along the y-axis of the subsection of the composite image highlighted by the white frame in (Fig. 5a). The profiles clearly show that the periodic artifact in the raw images is strongly reduced by the correction methods: the restored images show reduced intensity drop at the edges of neighbouring tiles and more uniform average level within single tiles. The beneficial effect of EVEN is marked by white frames in Fig. 5d. The correction methods reveal border structures that were faintly apparent in the raw image. However, depending on the processing method, single channels might be enhanced differently, resulting in a variable tone in the multicolour image. In this scenario, EVEN ensures the selection of channels with the top global quality features. In the Cellpose segmentation, EVEN optimization enables the identification of more cells and with better outline. Supplementary Fig. 29 shows intensity profiles extracted from a single tile, with further investigation of the colour differences. Supplementary information shows the optimization of further experimental datasets, including the comparison of different experimental settings for the measurement of stained cells (Supplementary Fig. 30) and quantification of cell segmentation (Supplementary Fig. 31). Supplementary Fig. 32 shows the mitigation of colour distortion issues caused by the possible non-consistent performance of single methods for different channels. Supplementary Fig. 33 shows the generation of quality rankings for single-channel measurements of a mouse brain slice, and Supplementary Fig. 34 shows the flexible adaptability of EVEN to the optimization of BaSiC hyperparameters for a timelapse movie with lateral uneven illumination and temporal drifts in the baseline intensity.

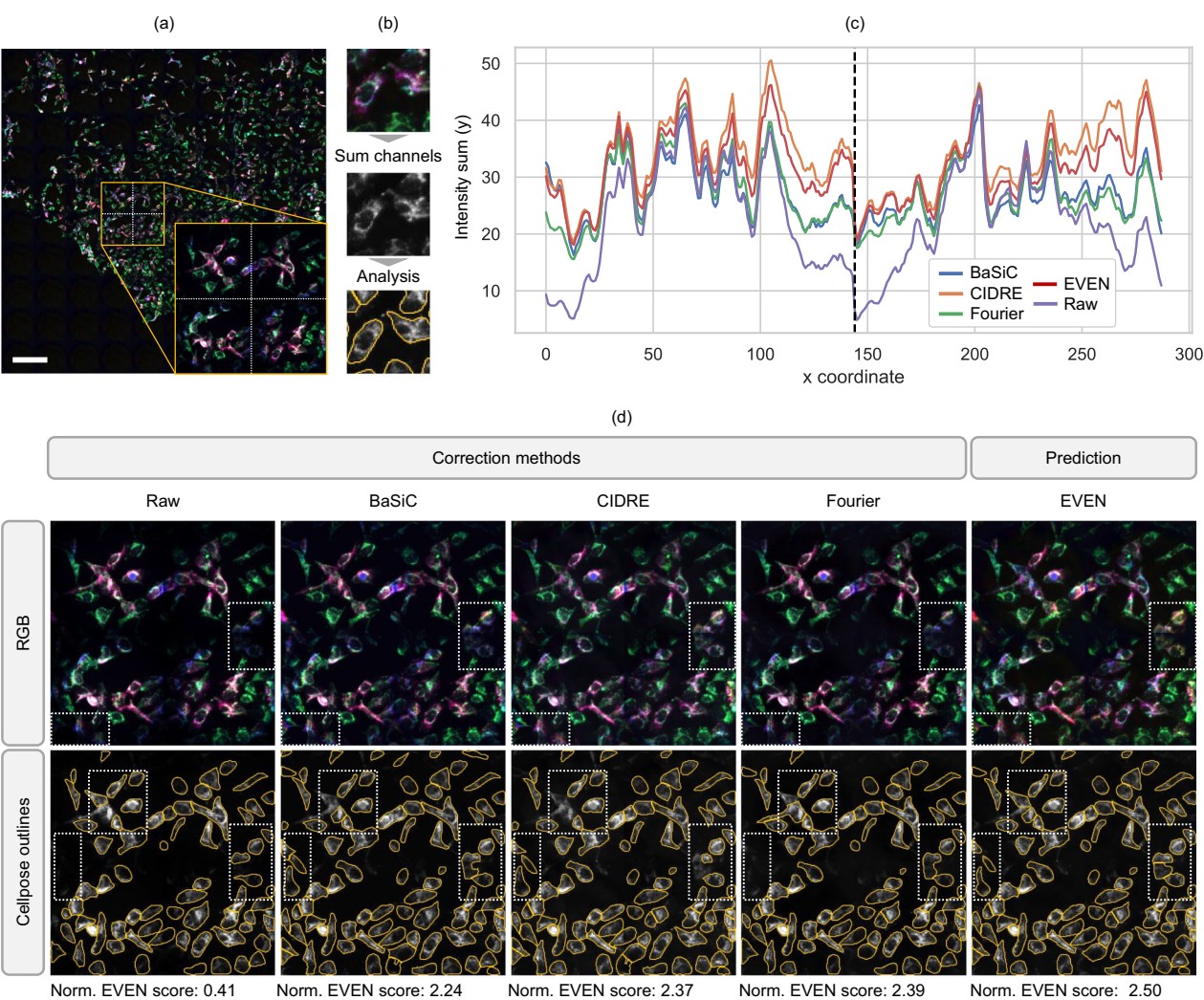

**Fig. 5 | Evaluation and enhancement of experimental measurements of stained cells.** A three-channel fluorescence microscopy measurement of stained HEK293 cells measured by Ph2 objective is automatically optimized by EVEN (prediction dataset 2, red: peroxisomal proteins (anti-GFP nanobody); green: TOMM20 protein; blue: peroxisomal proteins (eGFP)). **a** Raw multi-channel image. The inset shows the 2 × 2 tile section of the image used in this figure, with dashed white lines marking tile borders. Multiple corrections are obtained by applying BaSiC, CIDRE, Fourier methods, and then optimizing the multi-channel image with EVEN. EVEN selects CIDRE for the red and green channel, and Fourier for the blue channel. **b** Steps to analyse the measurements of stained cells: multi-channel images are converted to greyscale by summing the single channels (that contain signals from different components of the cytoplasm) and are analysed with automatic cells segmentation using Cellpose[19]. The greyscale image is obtained for the raw measurement, the single-channel corrections and the EVEN optimization. **c** Intensity sum (along y) of the greyscale inset for each method. The black dashed line indicates the border between neighbouring tiles. The corrected images show higher intensities at the edges of the tiles and the enhancement of sample features. EVEN and CIDRE show the greatest intensity recovery between tiles. **d** Top row: multi-channel images obtained with single-method corrections and EVEN optimization; the white dashed boxes highlight two regions significantly improved by EVEN. Bottom row: Cellpose prediction on the greyscale sum of the three channels for each method. After correction of uneven illumination, Cellpose can outline a greater number of cells, especially at the borders of neighbouring tiles. White dashed boxes highlight three regions where EVEN optimization provides better identification of the cells compared to non-optimized images. Bottom labels show, for each image, the normalized EVEN score summed over three channels and the cell count in the zoomed region. While counts are not strictly correlated with segmentation performance, good correction of uneven illumination enhances downstream analysis and generally increases the number of detected cells. Further quantification is provided in Supplementary Fig. 31. Scale bar: 180 μm, size of a single tile.

## Discussion

In this manuscript, we presented a method, named EVEN, for the automatic evaluation and enhancement of images affected by uneven illumination. The method is based on the parallel evaluation and comparison of multiple corrections of the same sample, and it has been implemented by following three main steps (Fig. 2). First, we identified the two main effects of uneven illumination on measured images, i.e., vignetting and the mosaicking effect, and we defined and assessed markers to track these artifacts, namely the edge energy ratio and the positive prominence. In the second step, we automated the evaluation process by utilizing the metrics to train an LDA model to differentiate between images with and without uneven illumination.

We demonstrated that the binary classification shows good agreement with visual perception of the images, and we validated the results on a prediction dataset. In addition, we built quality rankings from the model output thanks to a simple interpretation of the decision score. Finally, in the last step we applied EVEN to the quality enhancement and optimization of unseen multimodal datasets. EVEN can be easily applied to multimodal measurements to obtain an automatic and reliable optimization of single channels, that are merged in the final corrected image (Fig. 4, 5 and Supplementary Figs. 6–31, 35).

The results reported by this study open the way for further exploration and generalization of the method. The current manuscript was focused on the selection of the top images by analysing two quality

metrics and addressing features generated by uneven illumination only. However, multiple artifacts might be present in the images and the processing methods might introduce unwanted effects, such as smoothing, or fail to remove other negative features, such as noise. Therefore, a further exploration might include fine-tuning of additional quality metrics. This adjustment could be applied as an additional step, after prior selection of images with acceptable removal of uneven illumination, to determine the best compromise between flat-field correction and general image quality improvement.

One drawback of our method is currently the need of more than one available correction to generate optimized images, which cannot be determined a priori from the raw measurement: while the evaluation and prediction can be relatively fast, a large computational and time effort is required to obtain the output of the different correction methods for each channel (Supplementary Table 1). Indeed, the most reliable methods require the application of iterative processes, or the computation of the Fourier transform for composite images made of a large number of tiles. Further studies should focus on the automatic selection of the best method for an input image based on the features of the raw measurement, as the performance of the algorithms depends on the prior intensity distribution in the measurement and the number of tiles of the image to be corrected. However, when the optimal method cannot be determined beforehand, EVEN does not significantly increase computational time and can speed up the selection of the best result through automatic optimization. EVEN is a valuable choice for enhancing image analysis and the final interpretation of the measured sample if real-time corrections are not required.

EVEN is based on quantitative evaluation metrics and it can be integrated into any pipeline for the processing and analysis of optical microscopy measurements when an optimized correction of uneven illumination is needed, especially for optical systems, like fibres-based devices, that are particularly affected by uneven illumination (Fig. 1). Within the experimental pipeline, EVEN can be utilized to detect uneven illumination in raw images and tune the acquisition parameters to assess the quality of the acquired images (Supplementary Fig. 30). During image processing, EVEN can be utilized to fine tune parameters of algorithms to improve the performance of single methods. Finally, before image analysis, EVEN provides the best predicted correction by selecting for each channel the method that shows the optimized reduction of uneven illumination. Channel-based optimization balances the contribution of multiple colours in shaded regions, as uneven illumination varies with the detected signal due to sample, optics, and detector properties. Further research into how to determine reference colour balance for downstream tasks would further improve the optimization capabilities for uneven illumination.

EVEN requires minimal effort by the user, as it relies only on basic knowledge about the tile size for full automation. If the uneven illumination pattern is more complex than usual vignetting, the user can tune the quality markers and opt for a refined approach[20]. In addition, EVEN allows comparison of different corrections by using images with a few tiles per side and does not limit the evaluation for practical cases with limited number of tiles. To facilitate the implementation of our method and encourage the integration of EVEN in the experimental workflow, we provide all the essential tools in a public repository. These computational tools can be used to incorporate other markers into the method to suit other imaging experiments and transfer the method to any artifact type. For example, the edge energy ratio can be balanced in the horizontal and vertical directions to model a specific artifact more effectively, as shown in Supplementary Fig. 34. In this way, the EVEN method can be tailored. However, the pre-trained EVEN method presented here already works across a variety of samples, image modalities and artifacts, despite being trained only on tissue sections measured with multi-contrast microscopy.

## Methods

### Experimental datasets
Our study includes a variety of datasets acquired by different laboratories and with different techniques, including multimodal nonlinear imaging (CARS, TPEF, SHG) and fluorescence microscopy, measurements of tissues and cells.

**Manual assessment and model training.** The single-channel images used for the manual assessment of uneven illumination, the generation of semi-synthetic images and the model training are part of an experimental dataset introduced by the previous work of Chernavskaia et al[9]., composed of CARS, TPEF and SHG measurements of different biological specimens, including human head and neck tissue sections[13], human skin[21], human tissue biopsies from colonoscopy or surgical resection[22], mice colorectal biopsies[23], and pig brain tissue[18] (Supplementary Fig. 1). Each single-channel raw image is corrected by four different methods in the direct and spatial frequency domain. Further details on the images selected from this dataset are provided in the paragraph on model training.

**Prediction dataset 1: multimodal nonlinear measurements of human head and neck tissue.** The measurements predicted and optimized by EVEN are sections of 20 head and neck tissue samples taken during tumour surgery from 15 patients with head and neck cancer, introduced in a previous work[18]. After surgery, the samples were preserved by freezing in liquid nitrogen and then stored at −80 °C. 30 μm thick frozen sections were cut with a cryotome (Leica Biosystems GmbH, Wetzlar, Germany), then the slices were measured by an endomicroscopic system. The study was approved by the ethics committee of the University Hospital Jena (No. 4291-12/14). Written informed consent was obtained from each patient. The measurements were acquired by an endomicroscopic platform introduced in previous works[24]. The system includes different channels to measure CARS, SHG and TPEF signals, with excitation wavelengths and detection parameters tuned to collect relevant spectral regions for the extraction of morpho-chemical tissue information. The CARS signal is generated by a pump signal at 796 nm and a Stokes signal at 1030 nm, that match the $CH_2$ stretching vibrations abundant in lipids and proteins. With the same excitation scheme, TPEF emission can be collected from tissue autofluorescence generated by elastin, NAD(P)H, FAD, keratin, and collagen, and a SHG of the Stokes beam generated by the collagen fibres around 515 nm[9,21,25]. The acquisition process was executed with a custom-made software, based on the Scope Foundry python framework (http://www.scopefoundry.org/). The entire area of the specimen can reach a size of about 1 cm². To image the whole region of interest, the measurement was performed by raster scanning. Single tiles have a size of 430 μm × 430 μm, captured in an image of 1200 px × 1200 px, resulting in a pixel size of 360 nm × 360 nm. The pixel dwell time is 3 μs and each frame was averaged 5 times. Adjacent tiles have a spatial overlap of 10% of the tile size on each side. The raw channels were percentile normalized between 0 and 99 before processing. Then, they were corrected, and the tiles were re-stitched by removing 5% of the image pixels from each side. Further percentile normalization between 1.5 and 99 was applied to improve the visualization. Binning of 4 × 4 was applied before computation of the quality metrics to reduce the execution time. The original measurements are publicly available on Zenodo[26].

**Prediction dataset 2: multichannel measurements of stained HEK293 cells.** HEK293 cells (293 [HEK293] (ATCC CRL-1573), ATCC, USA) modified to express GFP on peroxisomal proteins were immunolabelled with FluoTag-Q anti-GFP nanobodies tagged with Alexa 647 (NanoTag Biotechnologies, Germany) at a dilution of 1:500. Additionally, immunolabeling was performed on TOMM20-protein with anti-Tomm20 rabbit polyclonal antibodies (proteintech, USA), dilution

1:200, and goat anti-rabbit IgG secondary antibodies labelled with Abberior STAR Orange (Abberior, Germany) at a dilution of 1:350. Measurements were performed on a Zeiss Elyra7 epifluorescence microscope equipped with a pco.edge sCMOS (version 4.2 CL HS) camera. The two sets of measurements are composed of three fluorescent channels (blue: eGFP, excitation at 488 nm; green: Abberior STAR Orange, excitation at 561 nm and red: AF647, excitation at 640 nm) and were acquired with an EC-Plan Neofluar 40x/0.75 Ph2 objective and a Zeiss Plan-Apochromat 40x/1.4 Oil DIC immersion objective. The FOV was maximized, resulting in multicolour images of size of 1028.79 μm x 1028.79 μm composed by $10 \times 10$ tiles of the same size, with a resolution of about 6.37 pixel/μm. Raw single channels were rescaled between 0 and 1 and processed with BaSiC, CIDRE and Fourier method. The output channels are percentile normalized between 0 and 99.9 to improve the visualization.

***Prediction dataset 3: multichannel measurement of cell culture[10].*** We applied EVEN to the three-channel measurement of cell culture provided by a previous work. The multicolour image is composed by DAPI, FITC and Cy3 fluorescence channels. We pre-processed the raw image and the BaSiC correction provided by the authors by percentile normalization between 0 and 99.99, and applied CIDRE and the Fourier method to the raw single-channel images.

***Prediction dataset 4: single-channel measurements of brain sections[10].*** We utilized the single-channel measurement of a mouse brain slice to generate an automatic quality ranking for different correction methods. The measurement was percentile normalized between 0 and 99.9 and corrected by CIDRE and Fourier before stitching. Then, raw and corrected images were predicted by EVEN and the final composite images were generated by applying the custom stitching protocol provided by the authors.

***Prediction dataset 5: timelapse movie of differentiating mouse hematopoietic stem cells[10].*** We applied EVEN to a timelapse movie of 100 frames imaging the differentiation of mouse hematopoietic stem cells. The images are characterized by lateral shading artifact on the right edge and temporal flashing due to changes in the microscopy settings during the measurement. We corrected the movie with BaSiC applying different hyperparameters: no temporal drift with automatic options, temporal drift with lambda flat parameter equal to 0, 0.5 (automatic), 2, 4, 6. Then, we built a composite image by stitching the frames in a $10 \times 10$ grid and we ranked the corrections using EVEN. Due to the lateral shading, we calculated a different energy ratio for this dataset, as explained below.

## Modelling of uneven illumination
To generate simulations of experimental images affected by uneven illumination, we follow the approach adopted by previous works[5,10]. Given the ideal measurement $t_0$ of a single tile, the vignetted FOV $t$ can be modelled as:

$$t(x, y) = t_0(x, y)M(x, y) + n(x, y) \quad (1)$$

Where $M(x, y)$ is a multiplicative illumination mask and $n(x, y)$ is an additive noise contribution. $M(x, y)$ is generated as a 2D Gaussian function maximum-normalized to 1, with an extension of the illumination mask tuned by the standard deviations $(\sigma_x, \sigma_y)$ along the two axes and the shift of the peak from the centre of the image tuned by the peak coordinates $(x_0, y_0)$. For a composite image $I = \sum_{i=1}^{n_t} t_i(x, y)$ made of $n_t$ tiles, we generate the mosaic effect by a composite illumination matrix consisting of equal illumination masks. The simulated dataset used for manual assessment is generated from 10 experimental single-channel images composed by multiple tiles. Uneven illumination is generated for each FOV with the

**Table 1 | Parameters for the generation of simulated uneven illumination**

| Property | Extension of uneven illumination | Position of maximum intensity |
|---|---|---|
| Free parameter | $\sigma_x, \sigma_y$ | $x_0, y_0$ |
| Value interval | [0.4, 1.2] | [0, 0.2] |
| Distribution | Constant increase in the interval | Random values with uniform distribution |

parameters reported in Table 1, that are multiplied by the lateral size of the square tile before the generation of the illumination mask. We generated 5 increasing levels of uneven illumination for each high-quality measurement, characterized by an illumination mask with constant decreasingly extension, random shift along x and y with respect to the centre of the FOV, and a low level of mixed Poisson-Gaussian noise.

## Image processing
Experimental images are corrected using three correction methods for uneven illumination available in the literature. The selected methods represent established and easily applicable tools, working in the direct and Fourier domain. This choice is only intended to validate EVEN's effectiveness and not to provide all the processing options, whose choice is much larger and in continuous development.

*CIDRE[5].* CIDRE is a spatial domain retrospective method that estimates both the illumination function and the additive term. CIDRE assumes that the measured objects may appear anywhere in the images with equal probability and that the underlying intensity distribution that generates uneven illumination is common to all measured FOV. We corrected the images with the Fiji implementation (v0.1) and using the default parameters provided in the plugin. The correction model is built for single channels and applied with zero-light preservation.

*BaSiC[10].* BaSiC is a spatial domain retrospective method inspired by CIDRE that estimates the illumination function and the additive term in a sequence of images by implementing a sparse and low-rank decomposition. We corrected the raw images using the Fiji implementation of BaSiC. The shading estimation is computed by estimating both the flat-field and dark-field shading profiles. The regularization parameters are automatically set, the temporal drift is ignored, and the lambda parameters are bot set to the automatic option of 0.5. As for CIDRE, the correction model is independently estimated for each channel.

*Fourier method.* We implemented a Fourier transform-based technique following the approach of a previous study[9,12]. This method is applied in the frequency domain, and it is based on the fact that a periodic mosaic artifact in the composite image generates a series of periodic peaks in the frequency domain. The suppression of these periodic peaks, whose position can be easily determined, can be utilized to eliminate the uneven illumination artifact.

*Processing workflow.* For each raw composite image, we followed the same workflow and applied the available correction methods to obtain different outputs for each measurement. The raw images are always rescaled to values between 0 and 1. If required, percentile normalization is applied to remove few bright pixels and improve the dynamic range of the measurements. Then, the three correction methods are applied, and the tiles are re-stitched after correction if neighbouring tiles have an overlapping region. Further normalization might be applied for an improved visualization of the raw and corrected images.

## EVEN: automatic optimization of the corrections
*General workflow.* The implementations of EVEN and the results are described by splitting the workflow in three main sections (Fig. 2).

EVEN has basic requirements for the images to be processed: it should be applied to stitched images with a minimal number of tiles of equal size for the correct computation of the metrics. The evaluation can be executed before the final stitching if the generation of the composite image affects the regular periodicity. EVEN's performance might be impacted by shading artifacts with limited periodicity, such as irregular stripes and gradually varying shading. The pre-trained model shows good performance on a variety of images and, if needed, the software tool allows to tune the set and definition of the quality metrics to match more complex artifact patterns.

*Step 1: quality metrics for image evaluation.* Two quality metrics are selected to detect the presence of vignetting in single tiles and the presence of the mosaic artifact in composite images for a composite image of size $N \times M$ pixels and composed by square tiles of $T \times T$ pixels.

- *Edge energy ratio.* The first metric is defined to detect shading in single tiles, i.e., the decrease of intensity in specific regions of the field-of-view. Since composite images are affected by shading in each tile $t_i$, we quantify the overall effect on the sum of tiles $t_{sum} = \sum t_i$. A simple but effective way to quantify the intensity decay in a region of the image is to define a ratio between the pixel energy in the shaded region $E_{shaded}$ and the total energy of the image $E_{tot}$:

$$E_{edge} = \frac{E_{shaded}}{E_{tot}} = \frac{\sum_{x_i, y_i \in S} t_{sum}(x_i, y_i)}{\sum_{x_i, y_i} t_{sum}(x_i, y_i)} \qquad (2)$$

The shaded region $S$ is defined depending on the uneven illumination in the current set of measurements. One of the most common shading effects is vignetting, which is a radial decay from the optical axis to the edges. In this case, the shaded region is defined by a radial threshold that splits the well illuminated centre of the image from the shadowed borders:

$$E_{edge} = \frac{\sum_{\sqrt{x_i^2 + y_i^2} > k_{rad}} t_{sum}(x_i, y_i)}{\sum_{x_i, y_i} t_{sum}(x_i, y_i)} \qquad (3)$$

Where $(x_i, y_i)$ are the coordinates of the *i-th* pixel of $t_{sum}$, and $k_{rad}$ is a radial threshold that identifies the edges of the image. A general rule to select the threshold is to set $k_{rad} = T/\sqrt{2\pi}$ and split $t_{sum}$ into two regions of equal area. In this manuscript, we selected a similar value $k_{rad} = 3/8$ the tile size, meaning that 3/4 of the lateral tile size includes flat illumination. The exact value of the threshold is not strictly binding if the resulting ratio decreases with increasing vignetting.

In the case of non-radial vignetting, a different threshold can be set for $E_{shaded}$. For example, for a lateral shading on the left side of the tiles, we can define $E_{shaded} = \sum_{x_i > k_v} t_{sum}(x_i, y_i)$, by selecting all pixels with coordinate x larger than a vertical threshold $k_v$, which can be set equal to half the lateral tile size T. The obtained edge energy follows the same trend of the radial threshold and does not require further training with an additional dataset.

In this manuscript, we have used the radial definition for all datasets, except for prediction dataset 5, which required the vertical threshold. The edge energy can be computed on any number of tiles.

- *Sum of the prominence of periodic peaks in the power spectrum.* The second parameter characterizes the strength of the mosaic artifact in a composite image. The periodic grid in the direct domain generates a series of periodic peaks at specific frequencies in the spatial frequency domain. We compute the positive prominence of these peaks over the baseline in the power spectrum of the image. For a composite image of size $N \times M$ pixels and composed by square tiles of $T \times T$ pixels, the mosaic has a

period $T$, and $T$ peaks are generated in the power spectrum located at frequencies $f_x = \pm \frac{k_x N}{T}, f_y = \pm \frac{k_y M}{T}$ for $k = 1, \ldots, \frac{T}{2}$. Therefore, the overall positive prominence of a composite image $I(x, y)$ is the sum of the positive prominence of all the peaks in the power spectrum $P(f_x, f_y) = |\mathscr{F}(I)|^2$. The prominence is computed for the sum-normalized power spectrum with the function *peak_prominences* of the *SciPy* python library and the central peak at $k_x = k_y = 0$ is excluded. The sum normalization ensures that the peak prominence represents the fraction of image energy located at the frequencies related to the mosaic and decreases the metric variability across different image sizes. The peak prominence can be computed on a small composite image but requires few tiles for the generation of a basic grid (see Supplementary Fig. 35). The open-source repository includes also the sum of negative prominence and magnitude prominence for further exploration.

Image values are always rescaled between 0 and 1 before the computation of the metrics. Further details about the implementation of the metrics are provided in the public repository (https://git.photonicdata.science/elena.corbetta/even).

*Step 2.1: machine learning-based classification by Linear Discriminant Analysis.*

- *Training.* The training dataset for binary classification is composed of 40 good and 40 bad experimental images from the previous study of Chernavskaia et al[9]., which investigated different correction methods in the direct and spatial frequency domain and provided an expert evaluation of all results. In our training dataset, the bad images are experimental stitched single-channels CARS or TPEF measurements of biological specimens affected by uneven illumination. The good images are good corrections among the results provided by the study, selected by visual assessment and according to the visual evaluation labels provided by the previous authors. Some samples are included in both classes, but not all the images are paired. Supplementary Fig. 1 shows a subset of the training dataset, to demonstrate the high content and structural variability despite the limited number of images. The LDA model is trained on the quality metrics and the true label of the images. Quality metrics are maximum normalized before training. The consistency between the quality rankings of LDA and the alternative classifiers (Supplementary Fig. 3) is assessed using Kendall's rank correlation coefficient, calculated for each image and channel separately based on all decision scores, and then averaged over all rankings.

- *Test.* The performance of the model is assessed by 5-fold cross-validation. At each round, 20% of the images per class are used as test split and the model is trained on the remaining images. Balanced test subsets are selected to have each sample appearing only in the training or testing split.

- *Prediction.* The main prediction dataset is composed of 23 three-channel measurements of human head and neck tissue slices, measured with coherent anti-Stokes Raman scattering (CARS), second harmonic generation (SHG) and two-photon excitation fluorescence (TPEF). The images are composed by multiple tiles and measured with a point scanning approach by the endomicroscopic system, whose acquisitions are subject to a strong decrease of intensity while moving from the optical axis to the edges of the FOV. In addition, the multimodal and nonlinear nature of the measurements enhances the effects of uneven illumination. Each channel has been corrected by three different correction methods (BaSiC[10], CIDRE[5] and Fourier method[9,12]), and the quality metrics computed for single-channel images are maximum-normalized to the training dataset and given as input to the trained model to assign the input data to the good or bad class. To assess the performance of the model over the prediction dataset, we assign a visual evaluation to each single-channel image. The corrections

with an acceptable but not optimal correction result are labelled as 'good'.

*Step 2.2: interpretation of the model output and generation of quality rankings.* An in-depth automatic evaluation workflow can be generated by interpreting the model output. The classification rules of the LDA classifier are described by the decision functions $\delta_k(x)$ defined for every class $k$ and computed for the input sample $x$ to be classified. The decision functions are defined as linear combinations of the input features $x_f$ (i.e, the quality metrics computed for image $x$), with coefficients $c_{k,f}$, means $\mu_{k,f}$, and class priors $\pi_k$ that are determined by the training process[27]:

$$\delta_k(x) = \sum_{x_f} \left( x_f c_{k,f} - \frac{1}{2}\mu_{k,f} c_{k,f} \right) + \log \pi_k \quad (4)$$

The decision scores for the image to be predicted is calculated by inserting the quality metrics in Eq. (4). In case of a binary classification, the images are assigned to the good or bad class according to a single decision score returned by the LDA model that represents the difference $\delta_{\text{good}}(x) - \delta_{\text{bad}}(x)$. It results in a positive value for images assigned to the good class and a negative value otherwise. The weighting coefficients and means to define each class's decision function represent the relative importance given to each quality metric. A positive value of the coefficients is assigned to features whose value increases for high quality images, whereas a negative value is associated to features increased by the presence of uneven illumination, but they might have a complex behaviour to balance the effect of multiple features. In the simple case of two quality metrics, as expected, the edge energy ratio is weighted with a positive coefficient, meaning that an increase of this metric is related to a probability increase for the good class, and the opposite applies to positive prominence. In addition, the mean of the bad class is characterized by low edge energy and high positive prominence, while the mean of the good class shows opposite behaviour (Supplementary Fig. 2). When multiple corrections of the same image are available, the decision score can be utilized to select the correction with highest quality. Indeed, a higher decision score is related to higher probability assigned to the high-quality class by the model. Therefore, the relative value of the decision functions computed for different corrections of the same image can be used to build a quality ranking.

*Step 3: generation of optimized multimodal images with EVEN.* The workflow to optimize the correction of multimodal images is built on the interpretation of the predicted decision score as a quality score. For each measurement, a decision score is predicted for each correction available for the single channels. The best correction of each channel is selected as the one with highest decision score, and the optimized multichannel image is generated by merging the selected corrections.

## Image analysis: cell segmentation
We applied automatic cell segmentation with Cellpose package[19] (v3.1.1.1) to the fluorescence measurements of stained HEK293 cells acquired by Ph2 objective with the pretrained Cyto3 model[28]. Cyto3 predicts cell segmentations on measurements of the cytoplasm. Since the measured images contain signal from different components located in the cytoplasm, we applied Cellpose on the greyscale sum of the three channels with input cell diameter of 20 px and binning 8×8. The sum provides better segmentation than single-channel thanks to complementary signal from different components and allows a comparison between the EVEN optimization and the single-method corrections. The normalized EVEN score is computed by z-score normalizing the decision score of single-channel images over the dataset and summing the decision score of single channels for every multi-channel image.

## Processing time
The execution time of all steps is estimated for the multimodal image of Fig. 4 and is reported in Supplementary Table 1, including the average time of 5 processing attempts. Computations were performed on a Windows 10 Pro workstation equipped with an AMD Ryzen Threadripper 3960 × 24-Core Processor, 128 GB RAM, and an NVIDIA GeForce RTX 3090. The image size is 8400 × 13,200 px before stitching and 7560 × 11,800 px after stitching. EVEN evaluation includes the reading time of the images and computation of the quality metrics, EVEN optimization is the time required to load the selected channels and merge them into a multimodal image.

## Reporting summary
Further information on research design is available in the Nature Portfolio Reporting Summary linked to this article.

## Data availability
The data generated in this study have been deposited in the Zenodo database under accession code 10.5281/zenodo.14748299 [https://zenodo.org/uploads/14748299][29]. The raw data of prediction dataset 1 have been generated by a previous study[18] and have been deposited in the Zenodo database under accession code 10.5281/zenodo.14604802 [https://zenodo.org/records/14604803]. The original data for the training dataset have been generated by a previous study[9] and have been deposited in the Zenodo database under accession code 10.5281/zenodo.17642604 [https://zenodo.org/records/17642604].

## Code availability
The source code for the EVEN method is publicly available in the GitLab repository https://git.photonicdata.science/elena.corbetta/even with a small test dataset.

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

## Acknowledgements

This work is supported by the BMFTR, funding program Photonics Research Germany (13N15706 (LPI-BT2-FSU), 13N15719 (LPI-BT5), J.P. and T.B., 13N15713 / 13N15717, C.E.) and is integrated into the Leibniz Center for Photonics in Infection Research (LPI). The LPI initiated by Leibniz-IPHT, Leibniz-HKI, Friedrich Schiller University Jena and Jena University Hospital is part of the BMBF national roadmap for research infrastructures. The work presented has received funding from the European Union's Horizon 2020 research and innovation programme under grant agreement No. 860185 (PHAST), B.M., O.G.L., M.S., and J.P. Co-funded by the European Union (ERC, STAIN-IT, 101088997, TB). Views and opinions expressed are however those of the author(s) only and do not necessarily reflect those of the European Union or the European Research Council. Neither the European Union nor the granting authority can be held responsible for them. We thank the Microverse Imaging Center for providing microscope facility support for data acquisition. ELYRA 7 (used for producing Fig. 5, Supplementary Fig. 29, 30, 31) was funded by the Free State of Thuringia with grant number 2019 FGI 0003. The Microverse Imaging Center is funded by the Deutsche Forschungsgemeinschaft (DFG, German Research Foundation) under Germany´s Excellence Strategy – EXC 2051 – Project-ID 390713860, C.E. We thank Delgir Zakinova (Institute of Applied Optics and Biophysics, Friedrich Schiller University Jena), who prepared the samples of HEK293 cells used to generate Prediction dataset 2. C.E. and P.T. further acknowledge funding by the DFG (project number 316213987 – SFB 1278; GRK M-M-M: GRK 2723/1 – 2023 – ID 44711651; instrument funding ID 460889961) and the innovation program by the German BMWi (ZIM; project 16KN070967 / Lab-on-a-chip SMARTIES).

## Author contributions

Development and implementation of EVEN: E.C. and T.B. Measurement and data curation of prediction dataset 1: M.C., H.B., T.M.-Z., B.M., O.G.-L., M.S., and J.P. Measurement and data curation of prediction dataset 2: P.T. and C.E. Image processing and analysis: E.C., M.C., P.T., H.B., and T.B. Software: E.C. and T.B. Visualization: E.C. and T.B. Project administration: T.M.-Z., B.M., O.G.-L., M.S., C.E., J.P., and T.B. Supervision: T.M.-Z., B.M., O.G.-L., M.S., C.E., J.P., and T.B. Writing original draft: E.C. and T.B. All authors revised and edited the manuscript. All authors approved the final draft.

## Funding

## Competing interests

The authors declare no competing interests.
