## [Transparent Peer Review file · Nature Communications]

Automatic optimization of flat-field corrections by evaluation and enhancement (EVEN) in multimodal optical microscopy

Corresponding Author: Professor Thomas Bocklitz

Version 0:

Reviewer comments:

Reviewer #1

(Remarks to the Author)

This manuscript presents a novel machine learning-based method, named EVEN (Evaluation and Enhancement), designed to assess and optimize corrections for uneven illumination in optical microscopy. Uneven illumination is a significant challenge in microscopy, leading to artifacts that can distort image analysis. The authors propose a three-step approach: Quantitative evaluation of vignetting and mosaic artifacts using defined metrics. Automation through a Linear Discriminant Analysis (LDA) model, trained to classify images with and without illumination artifacts. Optimization of multimodal images by selecting the best correction method for each channel. The method is tested on human head and neck tissue samples and stained cell images, demonstrating its effectiveness in improving image quality. The availability of the code and datasets in public repositories enhances transparency and facilitates adoption by other researchers.

However, the manuscript has some areas that need improvement and issues that need to be addressed.

The explanation of the LDA-based classification model could be expanded, particularly regarding the feature selection process and its impact on the decision score.

It would be beneficial to include a brief discussion on alternative machine learning models and why LDA was chosen over other approaches (e.g., SVM, Random Forest, or Neural Networks).

A comparison of EVEN's effectiveness against state-of-the-art deep learning-based image enhancement methods would strengthen the claims. Need to compare with deep learning based methods. For example, the methods of refs 14 and 15.

The experimental dataset used in the paper is too limited and should include data from multiple aspects and machines.

Are there color distortion issues in image fusion using three different correction methods?

Using edge energy and spectrum to characterize image quality, without considering human visual perception, should be compared with other image quality evaluation indicators.

How to define the edge of image? In the formula, the edge energy needs to be explained in depth.

In Figure 5, there is not much difference in the correction results of different methods. Some figures (e.g., Fig. 3, Fig. 5) contain detailed visual comparisons, but may benefit from additional annotations to highlight key differences.

In the supplementary materials, it can be seen that the proposed method has produced some relatively poor results.

The reference is marking with confusion. The last reference is unclear. For the reference 12, there is no paper title.

(Remarks on code availability)

Reviewer #2

(Remarks to the Author)

Summary:

Uneven illumination is a well-known issue affecting nearly all microscopy images, particularly fluorescence microscopy, where image intensity is crucial for quantification. Several existing methods address this issue; however, the correction quality for test image samples typically relies on subjective assessments by biologists or data owners, lacking an objective measurement. Motivated by this issue, the authors propose a machine learning-based method named EVEN to objectively

evaluate corrections by three existing methods—Cidre, BaSiC, and Fourier—and select the optimal correction for each fluorescence channel. EVEN is based on linear discriminant analysis (LDA), trained using 40 visually classified 'good' and 'bad' single-channel stitched images. The model was tested on 23 three-channel stitched fluorescence images of stained head-and-neck tissues.

The motivation behind the study is compelling, as objective quality assessment of illumination corrections remains challenging in practice, typically conducted case-by-case. Moreover, some correction methods involve non-trivial parameter tuning, making optimal parameter selection difficult for biologists. This paper provides a valuable step toward objective assessment. Despite the strong motivation, I believe the study does not fully resolve the issue due to several factors: lack of true ground-truth labels ('good'/'bad' images are visually determined, which may introduce bias), relatively low accuracy of the trained LDA model, limited model generalizability, and potentially suboptimal parameter selection for the correction methods (see Major Issues). Furthermore, the title and abstract refer to 'multi-modal' images, whereas only fluorescence images (single modality, multi-channel) are discussed. It would be more accurate and conventional to use the term "multi-channel."

Major Issues:

Training Images and Labels: The 'good' and 'bad' images used to train the LDA model were visually determined. Presumably, 'bad' images represent non-corrected stitched images, while 'good' images represent corrected ones. It is unclear which correction methods (Cidre, BaSiC, Fourier, or all three) were used to generate 'good' images and whether these selections were made by an expert. To reduce subjectivity, employing multiple experts or performing multi-expert voting is recommended.

LDA Model Performance: The reported balanced accuracy (0.73), sensitivity (0.84), and specificity (0.62) of the LDA model indicate relatively low performance compared to standard machine learning approaches. Approximately 70% agreement with expert assessments suggests limited reliability. Potential causes include noisy labels due to subjective visual assessments and limitations inherent in linear modeling. Exploring non-linear models such as Random Forest or Quadratic Discriminant Analysis (QDA) might significantly improve accuracy.

Model Generalizability: The LDA model was trained on a limited dataset comprising 40 single-channel images from a single tissue type (head-and-neck) and evaluated on just 23 three-channel images from the same tissue type and lab. This raises concerns about the model's generalizability. How consistent are the model's predictions when applied to different tissue types or images from other labs?

Evaluation Metrics: In addition to the LDA score, two metrics were proposed: intensity-based E_{edge} and spectrum-based P_{+} . E_{edge} assumes homogeneous intensity distribution, which might not hold true in many practical cases. For example, when few tiles are present, edge tiles significantly influence the metric, reducing its validity. Intensity variations within tissue foreground (e.g., vessels) could further affect the metric. Why not use overlap differences from stitched regions, as done in the original CIDRE and BaSiC papers?

The metric P_{+} assesses periodic grid artifacts and can only evaluate stitched images. However, it might be inherently biased towards Fourier-based methods, as both the metric and Fourier method operate directly in the frequency domain.

Assessing correction performance remains challenging. Showing improvements in downstream tasks, such as cell segmentation or biologically relevant quantification, could strengthen the paper significantly.

Parameter Tuning: The authors propose using EVEN for tuning correction parameters, such as in BaSiC. Although promising, the study fixes the smoothness regularization parameters for flatfield and darkfield (λ_{flat} and λ_{dark}) at 0.5 without exploring other values. It would be beneficial to test different parameter settings, including turning off the darkfield correction and using flatfield correction alone.

Processing Time: The processing time of EVEN compared to the original three correction methods should be reported to assess practical utility.

Minor Issues:

The pie charts in Fig. 3f and 5h are derived from different data types (FACS vs. time-lapse analysis) but presented similarly. Using different presentations to clearly differentiate these underlying data types is suggested.

Labelling inconsistencies (doc/docetaxel, control/ctrl, parental in Fig. 3b, and concentration of Docetaxel in Fig. 5a) should be addressed for consistency and clarity.

(Remarks on code availability)

I only took a very quick look at the code, which seems ok to me. In the python notebook, it uses BaSiCPy but in the paper it is stated use BaSiC Fiji version. Maybe authors can add a comment to clarify this confusion.

Reviewer #3

(Remarks to the Author)

In the manuscript by Corbetta et al. entitled "Making uneven illumination EVEN: automatic Evaluation and Enhancement of vignetting corrections in multimodal images", authors propose a machine-learning based method, named EVEN, to assess and optimize corrections of uneven illumination in optical microscopy. The uneven illumination is a very important issue in optical microscopy, and some results in this study are promising. However, key method details are missed and some important concerns need to be addressed.

Major concerns:

1. Since authors use established methods (BaSiC, CIDRE, and Fourier) for image correction, then the key novelty of this study relies on the two markers, the edge energy ratio and the positive prominence, and the general framework of multi-modal image processing. Therefore, I think the novelty of this study is limited.
2. It is not clear why these three established methods are selected. Authors do not mention the rationale of choosing these methods, especially when there are lots of emerging methods in dealing with uneven illumination.
3. Authors mention that they built quality rankings from the model output thanks to a simple interpretation of the decision

score. However, it is not clear how they define and obtain the decision score, and how this score balances the two main effects of uneven illumination on measured images, i.e., vignetting and the mosaicking effect.

4. An important question that needs to be answered is that how will the correction affect subsequent analysis. As authors mentioned in the abstract, the uneven illumination would impede the image analysis. I believe that demonstration of how the image analysis would be improved following correction of uneven illumination, by additional experiment or analysis, would greatly strengthen this study.

Specific comments:

5. The title could be further improved. I wonder why authors mention "vignetting" only in the title.

6. I do not think that putting Fig. 1 in Introduction is proper. The authors tend to mention a general pipeline, while Fig. 1 is specific to EVEN (the method proposed in this study). Rather, I think it fits to Results better.

7. I do not quite understand Fig. 3c. How do you select "Good" and "Bad" data? Why do authors write "visual evaluation" in the figure? These concerns also apply for the analysis of the positive prominence metric (Fig. 3d-f). Authors are suggested to provide the context.

8. Is the positive prominence metric dependent on other properties of the image, such as the size? Also, do authors sum all the prominences to acquire this metric?

9. I highly recommend that authors re-organize some contents. For example, in Fig. 3c and Fig. 3f, they show the statistical difference of the metrics computed for 40 experimental images affected by strong uneven illumination and 40 experimental images that were well corrected by processing methods; however, they have not yet detailed how they perform the image enhancement/correction until Fig. 4.

10. It looks that the sample size for LDA training is quite limited, which might impact the accuracy of the model.

11. Page 7, Paragraph 3, it should be Fig. 2(c).

12. In Fig. 4, authors mention that Fourier results in the optimized correction for CARS. However, it is revealed from the resultant image that a lot of details within the CARS image are missed following Fourier processing, which might be a reason of the reduction in mosaic effect. Please clarify this.

13. In Page 8, I wonder why authors mention "visually equivalent", since they have quantitative metrics to evaluate the corrected images.

14. It is good that authors show a huge number of examples in Supplementary Figures (Fig. S4-Fig. S26). However, for some raw images, I think they themselves are good enough (e.g., Fig. S25), and the optimization would not contribute to a better analysis of these images.

15. Authors mention that "Therefore, a fine-tuned optimization of the set of quality metrics might be required in case of unusual behaviour of the correction methods"; however, they did not propose how to do that and whether it will make sense. Authors are suggested to prove that.

16. In Fig. 5, to be honest, I could hardly tell which is better visually among EVEN prediction, BaSiC, and CIDRE, and more importantly, I do not believe that the subtle difference among these methods would really affect the subsequent analysis.

17. In Fig. 5, authors should explain what the three channels stand for.

18. In Fig. 5b, authors mention that "The profiles clearly show that the periodic artifact in the raw images is strongly reduced by the correction methods". However, although improved a little bit, the periodic artifact is still quite obvious as observed from these profiles, for all the three correction methods.

19. In page 8, authors mention: "However, depending on the processing method, single channels might be enhanced differently, resulting in a variable tone in the multi-colour image. In this scenario, EVEN ensures the selection of channels with the top global quality features." I appreciate this; however, I wonder what is the ground truth which can be used to evaluate the correction.

(Remarks on code availability)

I think that the the code is a usable resource for the community.

Version 1:

Reviewer comments:

Reviewer #2

(Remarks to the Author)

The revision has addressed most of my previous comments. Only two relevant comments:

1. Classification accuracy vs ranking: I understand the authors rebuttal that the trained model is used for ranking, does not have to achieve 100% classification accuracy. Authors also add classification performance of other classifier (random forest, etc). How about check the consistency of different ranking using different classifier?

2. Reviewers add down-stream analysis like cellpose segmentation, which is nice and an important aspect. How about put some of quantification of cell segmentation (now Supplementary Fig. 5) into the main Fig. 4?

(Remarks on code availability)

Looks good to me now

Reviewer #3

(Remarks to the Author)

I am satisfied with authors's response to my comments and appreciate the revisions. I do not have any other concerns.

(Remarks on code availability)

I think that the the code is a usable resource for the community.

We thank the Reviewers for accurately summarizing the content of our manuscript, acknowledging the importance of the available code, and giving us the opportunity to improve the quality of our study. In the following, we address all the concerns raised by the Reviewers to strengthen the conclusions of our study.

We have replied to all questions raised by the Reviewers. We include in **black** the comments of the Reviewers, in **blue** our response and in **red** the modifications applied to the manuscript. All the modifications are highlighted in **red** also in the manuscript and in the supporting file.

For an increased readability of the responses, we have included only the major modifications to the manuscript and Supplementary information. Every modification reported here is implicitly recalled and described in the main text, where the reference is highlighted in **red**.

We have also applied minor modifications to adapt the manuscript to the formatting guidelines of Nature Communications.

Reviewer #1 (Remarks to the Author):

This manuscript presents a novel machine learning-based method, named EVEN (Evaluation and Enhancement), designed to assess and optimize corrections for uneven illumination in optical microscopy. Uneven illumination is a significant challenge in microscopy, leading to artifacts that can distort image analysis. The authors propose a three-step approach: Quantitative evaluation of vignetting and mosaic artifacts using defined metrics. Automation through a Linear Discriminant Analysis (LDA) model, trained to classify images with and without illumination artifacts. Optimization of multimodal images by selecting the best correction method for each channel. The method is tested on human head and neck tissue samples and stained cell images, demonstrating its effectiveness in improving image quality. The availability of the code and datasets in public repositories enhances transparency and facilitates adoption by other researchers.

However, the manuscript has some areas that need improvement and issues that need to be addressed.

1. The explanation of the LDA-based classification model could be expanded, particularly regarding the feature selection process and its impact on the decision score.

It would be beneficial to include a brief discussion on alternative machine learning models and why LDA was chosen over other approaches (e.g., SVM, Random Forest, or Neural Networks).

We thank the Reviewer for the comment, and we agree on the need of an in-depth description of the classification model for a good transparency of our method. We have modified the Result section to clarify the advantages of LDA in our case:

LDA is a simple and efficient approach known for low risk of overfitting, ideal to obtain a generalizable model when the training dataset is limited. Moreover, LDA classifies the images by generating an interpretable decision score, which is the linear combination of the input features weighted by class coefficients and means (Supplementary Fig. 2). The score can be used for the generation of quality rankings (see the Methods section).

Moreover, the Methods section includes now a detailed description of the generation of the decision score by LDA and clarified its dependency on the quality metrics:

The classification rules of the LDA classifier are described by the decision functions $\delta_k(x)$ defined for every class k and computed for the input sample x to be classified. The decision functions are defined as linear combinations of the input features x_f (i.e, the quality metrics computed for image x), with coefficients $c_{k,f}$, means $\mu_{k,f}$, and class priors π_k that are determined by the training process:

$$\delta_k(x) = \sum_{x_f} \left(x_f c_{k,f} - \frac{1}{2} \mu_{k,f} c_{k,f} \right) + \log \pi_k$$

The decision scores for the image to be predicted is calculated by inserting the quality metrics in the formula above. In case of a binary classification, the images are assigned to the good or bad class according to a single decision score returned by the LDA model that represents the difference $\delta_{good}(x) - \delta_{bad}(x)$, and it results to a positive value for images assigned to the good class and a negative value otherwise. The weighting coefficients and means to define each class's decision function represent the relative importance given to each quality metric. A positive value of the coefficients is assigned to features whose value increases for high quality images, whereas a negative value is associated to features increased by the presence of uneven illumination, but they might have a complex behaviour to balance the effect of multiple features. In the simple case of two quality metrics, as expected, the edge energy ratio is weighted with a positive coefficient, meaning that an increase of this metric is related to a probability increase for the good class, and the opposite applies to positive prominence. In addition, the mean of the bad class is characterized by low edge energy and high positive prominence, while the mean of the good class shows opposite behaviour (Supplementary Fig. 2).

We acknowledge the importance of a comparison of LDA with other established approaches. To this aim, we have compared LDA with Quadratic Discriminant Analysis (QDA), Linear Support Vector Machines (Linear SVM), Radial SVM and Random Forest Classifier (RFC). All methods have been trained on the same training dataset (composed of 40 good and 40 bad images) with 5-fold cross validation. The following plot shows that RFC is the only method with higher accuracy than LDA in the cross-validation process. However, it suffers from overfitting showing a lower accuracy once the final trained model is applied on the prediction dataset. We have added this comparison to the supplementary information and referenced it in the Results:

Supplementary Fig. 1 – Comparison between machine learning (ML)-based classifiers. 5-fold cross validation is executed with Linear Discriminant Analysis (LDA), Quadratic Discriminant Analysis (QDA), Linear Support Vector Machine (Linear SVM), radial SVM and Random Forest Classifier (RFC). (a) Mean accuracy for the same splits for each method. Random Forest (RFC) shows the best performance, followed by LDA. (b) Cross-validation accuracy averaged on the 5 splits. (c) Accuracy of the final

models on multimodal nonlinear microscopy measurements of head and neck tissue (prediction dataset 1). LDA keeps a good performance on the prediction dataset, while RFC shows overfitting.

2. A comparison of EVEN's effectiveness against state-of-the-art deep learning-based image enhancement methods would strengthen the claims. Need to compare with deep learning based methods. For example, the methods of refs 14 and 15.

We understand the Reviewer's request for a comparison with deep learning (DL) methods. Among the existing DL methods for flat field correction, we have tested the SSCOR method for striping artifacts (Ref. 15) because it is supported by an open-source repository and usable data.

Before presenting the results, we would like to emphasize that EVEN is not a direct correction method, like the DL approaches reported in the references, and it cannot be directly compared to DL correction. However, it can be used to select the best correction after striping removal.

Following the standard EVEN workflow, we applied SSCOR and other correction methods and evaluated the results with EVEN. As a preliminary test, we have retrained and applied SSCOR to the multimodal image shown in Fig. 4, following the instructions on the SSCOR repository. In this case SSCOR does not show a good performance than other methods and the EVEN optimization is not changed compared to the previous result, as described in Response Figure 1:

Response Figure 1 - Evaluation and Enhancement of multimodal nonlinear measurements of human head and neck tissue. One of the multimodal images of prediction dataset 1 is automatically optimized by EVEN. (a) The raw input measurement with strong uneven illumination. (b) The three channels are corrected independently with BaSiC, CIDRE, Fourier method, pretrained SSCOR and SSCOR retrained on the same image. The pretrained SSCOR model does not show an acceptable performance: the green and blue channels (showed with strong contrast adjustment in the single-channel visualization) have a low residual artifact but a consistent intensity loss; the red channel shows a strong residual artifact. The retrained SSCOR predicts well balanced colours, but strong residual uneven illumination. In this panel, we have applied contrast adjustment only to the single channels of pretrained SSCOR for a clearer visualization. (c) The EVEN optimization does not change compared to the result in Figure 4 of the manuscript.

We have then compared SSCOR against other correction methods using one of the multimodal images provided by Wang et al. (ref 15). SSCOR is a powerful method that can correct artifacts beyond the scope of our work, such as bubbles, irregular stripes and out-of-focus shading.

For a fair comparison with other methods, we have selected the two-channel stimulated Raman scattering (SRS) measurement of human brain tumour with simple with horizontal stripes. As expected, SSCOR is the best performing method from a visual point of view. However, after careful inspection of the raw measurement (Response Figure 2 (a)), we can observe that the shading artifact varies from the top to the bottom of the image. This reduced periodicity has an impact on the other correction methods, that remove an average uneven illumination. This results in a good correction in the centre of the image, a residual artifact at the top, and an overcompensation at the bottom. Moreover, the reduced periodicity impacts also the performance of EVEN, which cannot generate a comprehensive evaluation, although it shows an interpretable and acceptable result.

Response Figure 2 – Evaluation and Enhancement of SRS measurements of human brain tumors. (a) The raw input measurement with strong uneven illumination, composed by two channels (green: lipid, blue: protein). (b) Single channels are corrected independently with BaSiC, CIDRE, Fourier method, and SSCOR retrained on the same image. SSCOR shows the best performance, but the other methods are impacted by the non-perfect periodicity of the artifact. (c) Two-channel corrections and EVEN optimization. EVEN selects Fourier as best method, but its performance is also impacted by the reduced periodicity of the artifact. Nevertheless, EVEN produces an acceptable output.

We have underlined the potential of DL approaches in the introduction, emphasizing that their application can go beyond the standard uneven illumination discussed in our manuscript, and included a brief description of the suggested conditions for the optimal application of EVEN in the Methods, within the description of the general EVEN workflow.

3. The experimental dataset used in the paper is too limited and should include data from multiple aspects and machines.

We understand the Reviewer's concern about the size of the training dataset, and we would like to provide additional details about it to explain our reasoning.

The training dataset consists of single-channel images with variable content in terms of image brightness, sample type, and imaged structures. This was done to increase variability within the dataset despite the presence of the same artifact. To increase transparency, we have included a subset of the training images in the supplementary material:

Supplementary Fig. 2 – Example images from the training dataset. Subset of square crops from the training dataset, with tile size equal to 512 px, variable number of tiles and a wide variety of imaged structures. The first four images from the top right

are pared in the two classes. (a) Example images from the bad class: experimental uneven illumination. (b) Example images from the good class: good corrections of the experimental measurements.

We compared the performance of the model using a simulated dataset and did not find any significant advantages. Therefore, for our training, we decided to use completely experimental images to maintain the original experimental features of the artifacts.

We would like to emphasize that the training is not performed directly on the images, but rather on the quality features extracted from them. Therefore, our goal is to collect a set of images with varied content and a clear trend in quality features. As shown in Fig. 3 (c, f), the Bad and Good classes are well-separated. Furthermore, this trend is reflected in the coefficient values and means in Supplementary Fig. 2.

We have added a more in-depth description of the dataset in the Methods:

The single-channel images used for the manual assessment of uneven illumination, the generation of semisynthetic images and the model training are part of an experimental dataset introduced by the previous work of Chernavskaia et al.¹, composed of CARS, TPEF and SHG measurements of different biological specimens, including human head and neck tissue sections², human skin³, human tissue biopsies from colonoscopy or surgical resection⁴, mice colorectal biopsies⁵, and pig brain tissue⁶ (Supplementary Fig. 1).

And underlined the generalizability of EVEN in the Discussion:

These computational tools can be used to incorporate other markers into the method to suit other imaging experiments and transfer the method to any artifact type. For example, the edge energy ratio can be balanced in the horizontal and vertical directions to model a specific artifact more effectively, as shown in Supplementary Fig. 34. In this way, the EVEN method can be tailored. However, the pre-trained EVEN method presented here already works across a variety of samples, image modalities and artifacts, despite being trained only on tissue sections measured with multi-contrast microscopy.

4. Are there color distortion issues in image fusion using three different correction methods? We thank the Reviewer for highlighting the importance of multimodal measurements. To prevent color distortion issues caused by uneven illumination, we apply consistent normalization to the entire dataset for each image channel, both before and after correction and for all correction methods. Color distortions may be caused by the detectors or by the variable performance of the correction methods on different channels. There is no reason to assume that EVEN causes further degradation in this sense. In fact, EVEN selects the optimal channel for each scenario, preventing color distortions when a single method is not ideal for all channels.

We have specified this aspect in Supplementary Fig. 29 as follows:

The optimized image shows a stronger reduction of uneven illumination and a mitigation of possible colour distortion issues (such as the strong intensity reduction in the Cy3 channel caused by CIDRE) compared to the single-method cases.

And in the main text in the Results:

Supplementary information shows the optimization of further experimental datasets, including the comparison of different experimental settings for the measurement of stained cells (Supplementary Fig. 30) and the mitigation of colour distortion issues caused by the possible non-consistent performance of single methods for different channels (Supplementary Fig. 32).

And in the Discussion:

Channel-based optimization balances the contribution of multiple colours in shaded regions, as uneven illumination varies with the detected signal due to sample, optics, and detector properties. Further research into how to determine reference colour balance for downstream tasks would further improve the optimization capabilities for uneven illumination.

- Using edge energy and spectrum to characterize image quality, without considering human visual perception, should be compared with other image quality evaluation indicators.

We understand the Reviewer's concern about properly validating image quality. Image quality assessment (IQA) is a broad topic that includes various definitions of quality. In our study, we define image quality as an effective reduction of uneven illumination. We ignore other properties, such as resolution and noise content, because they are not typically affected by flat-field correction algorithms (Response Figure 3). Visual perception is used as a validation tool to demonstrate that edge energy and prominence in the power spectrum allow for an automatic ranking of images that aligns with an expert human evaluation.

The following plot of other established quality demonstrates that other quality aspects of the images are not affected:

Response Figure 3 – **Standard quality metrics computed for the training dataset, split in the two classes.** (a) Signal to noise ratio: ratio between the signal range of the image and the standard deviation of a background region. (b) Image resolution (in pixels) estimated by Fourier Ring Correlation (FRC). (c) Root mean squared contrast. (d) Structural complexity, computed as the average image gradient along the x and y direction. Opposite to the trend of the edge energy and peak prominence, these metrics do not show a relevant difference between good and bad class.

To underline the role of visual perception we have modified the text as follows:

Visual assessment from an expert evaluator can be valid, but not objective and quantitative, while the latter approach is time consuming and requires significant overlap between tiles; therefore, it is a

very effective approach for validation of new methods, but it is not practical in real-case experimental scenarios [...] Therefore, EVEN is a quality evaluation and optimization tool that makes the intensity distribution uniform from both a visual and quantitative, objective perspective.

How to define the edge of image? In the formula, the edge energy needs to be explained in depth.

We give high importance to the Reviewer's opinion about a correct and clear definition of the methods. We have formulated a more general definition of the edge energy and included it in the Methods section as follows:

Edge energy ratio. The first metric is defined to detect shading in single tiles, i.e., the decrease of intensity in specific regions of the field-of-view. Since composite images are affected by shading in each tile t_i , we quantify the overall effect on the sum of tiles $t_{sum} = \sum t_i$. A simple but effective way to quantify the intensity decay in a region of the image is to define a ratio between the pixel energy in the shaded region E_{shaded} and the total energy of the image E_{tot} :

$$E_{edge} = \frac{E_{shaded}}{E_{tot}} = \frac{\sum_{x_i, y_i \in S} t_{sum}(x, y)}{\sum_{x_i, y_i} t_{sum}(x, y)}$$

The shaded region S is defined depending on the uneven illumination in the current set of measurements. One of the most common shading effects is vignetting, which is a radial decay from the optical axis to the edges. In this case, the shaded region is defined by a radial threshold that splits the well illuminated centre of the image from the shadowed borders :

$$E_{edge} = \frac{\sum_{\sqrt{x_i^2 + y_i^2} > k_{rad}} t_{sum}(x_i, y_i)}{\sum_{x_i, y_i} t_{sum}(x_i, y_i)}$$

Where (x_i, y_i) are the coordinates of the i -th pixel of t_{sum} , and k_{rad} is a radial threshold that identifies the edges of the image. A general rule to select the threshold is to set $k_{rad} = T/\sqrt{2\pi}$ and split t_{sum} into two regions of equal area. In this manuscript, we selected a similar value $k_{rad} = 3/8$ the tile size, meaning that 3/4 of the lateral tile size includes flat illumination. The exact value of the threshold is not strictly binding if the resulting ratio decreases with increasing vignetting.

In the case of non-radial vignetting, a different threshold can be set for E_{shaded} . For example, for a lateral shading on the left side of the tiles, we can define $E_{shaded} = \sum_{x_i > k_v} t_{sum}(x_i, y_i)$, by selecting all pixels with coordinate x larger than a vertical threshold k_v , which can be set equal to half the lateral tile size T . The obtained edge energy follows the same trend of the radial threshold and does not require further training with a new dataset.

In this manuscript, we have used the radial definition for all datasets, except for prediction dataset 5, which required the vertical threshold. The edge energy can be computed on any number of tiles.

And we have modified Figure 3 to include a more in-depth description of the evaluation workflow:

Fig. 1 - Manual assessment of uneven illumination by quantitative metrics. (a) Workflow for calculating the edge energy ratio (E_{edge}) to detect vignetting in composite images: the image is split into single tiles, that are summed upon computation of the ratio. (b) Trend of E_{edge} for 10 semi-synthetic images with 5 increasing levels of uneven illumination. HQ is the high-quality image without any artifact. The shaded bar indicates the 95% confidence interval on the metrics computed for the 10 images with the same artifact level. (c) Value of E_{edge} for known experimental images (training dataset) with and without uneven illumination: a low value is correlated with the presence of strong artifacts. (d) Representation of the periodic peaks generated in the frequency domain by the mosaic artifact in composite images. The profile (orange dashed line) extracted from one of the main axes of the power spectrum, plotted in logarithmic scale, shows the positive prominence (P_+) of the peaks over the main baseline of the power spectrum (orange line in the red squared box, that represents the zoom of the highlighted region). (e) Trend of P_+ for 10 semi-synthetic images with 5 increasing levels of uneven illumination. HQ is the high-quality image without any artifact. The shaded bar indicates the 95% confidence interval on the metrics computed for the 10 images with the same artifact level. (c) Value of P_+ for known experimental images (training dataset) with and without uneven illumination: high prominence is correlated with the presence of strong artifacts.

6. In Figure 5, there is not much difference in the correction results of different methods. Some figures (e.g., Fig. 3, Fig. 5) contain detailed visual comparisons, but may benefit from additional annotations to highlight key differences.

We thank the Reviewer for the comment, and we acknowledge the importance of optimal visualization of our results. We have improved the visualization of the workflow in Figure 3 as explained before.

We have included intensity profiles in Figure 4 for a better quantification of the improvement provided by EVEN:

Fig. 2 - Evaluation and Enhancement of multimodal nonlinear measurements of human head and neck tissue. One of the multimodal images of *prediction dataset 1* is automatically optimized by EVEN. (a) The raw input measurement with strong uneven illumination. (b) The three channels are corrected independently with three different correction methods: BaSiC⁷, CIDRE⁸, Fourier¹. Due to the different intensity distribution in the channels, that are generated by the different morphological features of the different components of the tissue, the correction methods show a different performance for different channels. Indeed, the multimodal corrections obtained by single methods (bottom row) are different and show residual artifacts. (c) The optimized correction is predicted by EVEN after the computation of the quality metrics for each correction of the channels. For this specific measurement, the Fourier method is selected as the best correction for the CARS signal, as it provides flatter intensity distribution and enhancement of few bright regions without loss of details in the darker areas, whereas CIDRE is selected for TPEF and SHG channels. The optimized image shows a stronger reduction of uneven illumination compared to the single-method cases. (d) Intensity profile along *x* in the section of the image marked by the grey ruler. For each image, we generated a grayscale sum of the three channels and then plotted the maximum normalized intensity of the pixel sum along *y*. The red vertical lines mark the border of the tiles. EVEN shows the strongest reduction of vignetting at the tiles borders and the flattest profile. Scale bar: 200 μm .

In addition, we modified Figure 5 including a more straightforward visualization of the intensity profiles, now focused on a smaller region of the image, and white dashed boxes to highlight regions of interest in the multichannel images and greyscale analysed images:

Fig. 3 - Evaluation and Enhancement of experimental measurements of stained cells. A three-channel measurement of stained HEK293 cells measured by Ph2 objective is automatically optimized by EVEN (prediction dataset 2, red: peroxisomal proteins (GFP); green: TOMM20 protein; blue: peroxisomal proteins (anti-GFP nanobody)). (a) Raw multi-channel image. The inset shows the 2x2 tile section of the image used in this figure for further visualization of the results. Multiple corrections are obtained by applying BaSiC, CIDRE, Fourier methods, and then optimizing the multi-channel image with EVEN. EVEN selects CIDRE for the red and green channel, and Fourier for the blue channel. (b) Steps to analyse the measurements of stained cells: the multi-channel images are converted to greyscale (that contain signals from different components of the cytoplasm) and are analysed with automatic cells segmentation using Cellpose⁹. The greyscale image is obtained for the raw measurement, the single-channel corrections and the EVEN optimization. (c) Intensity sum (along y) of the greyscale inset for each method. The black dashed line indicates the border between neighbouring tiles. The corrected images show higher intensities at the edges of the tiles and the enhancement of sample features. EVEN and CIDRE show the greatest intensity recovery between tiles. (c) Top row: multi-channel images obtained with single-method corrections and EVEN optimization; the white dashed boxes highlight two regions significantly improved by EVEN. Bottom row: Cellpose prediction on the greyscale sum of the three channels for each method. After correction of uneven illumination, Cellpose can outline a greater number of cells, especially at the borders of neighbouring tiles. White dashed boxes highlights three regions where EVEN optimization provides better identification of the cells compared to non-optimized images. Scale bar: 180 μm , size of a single tile.

7. In the supplementary materials, it can be seen that the proposed method has produced some relatively poor results.

We value the Reviewer's comment, and we understand the concerns about few results. The Reviewer is right: some results in the supplementary material show room for improvement. However, most of these results are not due to EVEN's performance, but rather to the performance of the correction methods, which fail to fully eliminate the uneven illumination artifact.

EVEN is an evaluation and optimization method that only works within the boundaries set by the available corrections. Our method can optimize images by selecting the best available channels, but it cannot compensate for residual artifacts left by corrections.

When poor results are obtained as EVEN outputs, further optimization is needed before moving forward with the final analysis. However, for the purpose of our work, we believe it is beneficial to demonstrate EVEN's performance in the presence of suboptimal corrections to show that the results are consistent and represent the optimal compromise among the available options.

To clarify this point, we have improved the description of our results as follows:

When residual uneven illumination is present in all available corrections, the EVEN output is the best compromise among the available combinations (Supplementary Fig. 13, 14, 17). **Indeed, EVEN performs a good optimization within the limits set by the correction methods and cannot compensate for residual artifacts left by all methods. In this cases, further processing would be necessary to optimize visual interpretation and automatic analysis but for the demonstration of EVEN working principle it is beneficial to show the result also in presence of non-optimal images. In few cases (Supplementary Fig. 8, 12, 22, 27) all the corrections are visually equivalent and indeed most of their channels show a low variability in the decision scores of Supplementary Fig. 4 and quality metrics of Supplementary Fig. 5, demonstrating that EVEN do not generate unexpected results for high-quality images.**

8. The reference is marking with confusion. The last reference is unclear. For the reference 12, there is no paper title.

We thank the Reviewer for carefully checking the reference list. We have updated all references following the Journal's guidelines (improving the last reference to a dataset) and we have added the title to reference 12.

Reviewer #2 (Remarks to the Author):

Summary:

Uneven illumination is a well-known issue affecting nearly all microscopy images, particularly fluorescence microscopy, where image intensity is crucial for quantification. Several existing methods address this issue; however, the correction quality for test image samples typically relies on subjective assessments by biologists or data owners, lacking an objective measurement. Motivated by this issue, the authors propose a machine learning-based method named EVEN to objectively evaluate corrections by three existing methods—Cidre, BaSiC, and Fourier—and select the optimal correction for each fluorescence channel. EVEN is based on linear discriminant analysis (LDA), trained using 40 visually classified 'good' and 'bad' single-channel stitched images. The model was tested on 23 three-channel stitched fluorescence images of stained head-and-neck tissues. The motivation behind the study is compelling, as objective quality assessment of illumination corrections remains challenging in practice, typically conducted case-by-case. Moreover, some correction methods involve non-trivial parameter tuning, making optimal parameter selection difficult for biologists. This paper provides a valuable step toward objective assessment. Despite the strong motivation, I believe the study does not fully resolve the issue due to several factors: lack of true ground-truth labels ('good'/'bad' images are visually determined, which may introduce bias), relatively low accuracy of the trained LDA model, limited model generalizability, and potentially suboptimal parameter selection for the correction methods (see Major Issues). Furthermore, the title and abstract refer to 'multi-modal' images, whereas only fluorescence images (single modality, multi-channel) are discussed. It would be more accurate and conventional to use the term "multi-channel."

We thank the Reviewer for acknowledging the importance of the issues tackled by our study and the valuable contribution of our method. In the following, we answer to all questions raised by the Reviewer, and we do our best to solve the concerns and fill the gaps.

The term *multimodal image* refers to images generated by using different imaging modalities and can be used as long as the different measurements are characterized by signals generated by different processes and encode different information of the sample structure. The use of the term *multi-channel* does not imply necessarily the presence of complementary information and might be reductive in our case.

Our study investigates optical microscopy processes, including nonlinear techniques (such as coherent anti-Stokes Raman scattering, two-photon excitation fluorescence, second harmonic generation) and linear fluorescence microscopy. We recognize, however, that multi-channels is the correct term for some of the examples reported in the manuscript. Therefore, we have changed the name of the prediction dataset 2 and 3 from multimodal to multichannel and the respective references in the manuscript. Moreover, we have changed the title with the more specific term *multimodal optical microscopy*.

Major Issues:

1. Training Images and Labels: The 'good' and 'bad' images used to train the LDA model were visually determined. Presumably, 'bad' images represent non-corrected stitched images, while 'good' images represent corrected ones. It is unclear which correction methods (Cidre, BaSiC, Fourier, or all three) were used to generate 'good' images and whether these selections were made by an expert. To reduce subjectivity, employing multiple experts or performing multi-expert voting is recommended.

We thank the Reviewer for highlighting the need of more detailed information on the training dataset. The Reviewer's description of 'good' and 'bad' images is right, and we have supplemented the Methods section with more details as follows:

Manual assessment and model training. The single-channel images used for the manual assessment of uneven illumination, the generation of semisynthetic images and the model training are part of an experimental dataset introduced by the previous work of Chernavskaia et al.¹, composed of CARS, TPEF and SHG measurements of different biological specimens, including human head and neck tissue sections², human skin³, human tissue biopsies from colonoscopy or surgical resection⁴, mice colorectal biopsies⁵, and pig brain tissue⁶ (Supplementary Fig. 1). Each single-channel raw image is corrected by four different methods in the direct and spatial frequency domain. [...]

And:

Training. The training dataset for binary classification is composed of 40 good and 40 bad experimental images from the previous study of Chernavskaia et al.¹, which investigated different correction methods in the direct and spatial frequency domain and provided an expert evaluation of all results. In our training dataset, the bad images are experimental stitched single-channels CARS or TPEF measurements of biological specimens affected by uneven illumination. The good images are good corrections among the results provided by the study, selected by visual assessment and according to the visual evaluation labels provided by the previous authors. Some samples are included in both classes, but not all the images are paired. Supplementary Fig. 1 shows a subset of the training dataset, to demonstrate the high content and structural variability despite the limited number of images.

The correction methods included in the previous work include two spatial domain approaches (boundary adjustment method and histogram adjustment method), one frequency domain approach (Fourier), and the combination of Fourier with boundary adjustment method. We would like to emphasize that the full dataset is available from the previous manuscript and the source data is available in Zenodo. We investigated also the use of a simulated 'bad' images for the model training, where we have included variable uneven illumination to the good images. However, we have not found any relevant advantage in terms of performance, and we have therefore conducted our study with full experimental dataset.

2. LDA Model Performance: The reported balanced accuracy (0.73), sensitivity (0.84), and specificity (0.62) of the LDA model indicate relatively low performance compared to standard machine learning approaches. Approximately 70% agreement with expert assessments suggests limited reliability. Potential causes include noisy labels due to subjective visual assessments and limitations inherent in linear modeling. Exploring non-linear models such as Random Forest or Quadratic Discriminant Analysis (QDA) might significantly improve accuracy.

We understand the Reviewer's concern regarding the model classification accuracy, which results mainly because we estimate this value using a cross-validation approach and generate a (linear) LDA model including different tissue types and modalities in the training and validation data split. This means this performance is the model performance averaged over a variety of tissues as sample and modalities like CARS, TPEF and SHG to measure them.

Linear Discriminant Analysis (LDA) is an effective method in presence of limited datasets and provides good generalizability with limited risk of overfitting. Beyond these advantages, the main motivation to use LDA is its straightforward interpretation, which allows to easily build a quality score and an objective ranking of the images, as we have included in the Results. The limited classification performance is mainly caused by the large variety in the training dataset, which is composed of images acquired by different instruments and samples, and by the presence of artifacts with variable strength, which are difficult to locate in a strict binary classification. These aspects increase the classification complexity, but do not limit the generation of quality rankings.

We have compared LDA with Quadratic Discriminant Analysis (QDA), Linear Support Vector Machines (Linear SVM), Radial SVM and Random Forest Classifier (RFC). All methods have been trained on the same training dataset (composed of 40 good and 40 bad images) with 5-fold cross validation. The following plot shows that RFC is the only method with higher accuracy than LDA in the cross-validation process. However, it suffers from overfitting showing a lower accuracy once the final trained model is applied on the prediction dataset. We have included our findings in the supplementary material (and referenced them in the main text) as follows:

Supplementary Fig. 3 – Comparison between machine learning (ML)-based classifiers. 5-fold cross validation is executed with Linear Discriminant Analysis (LDA), Quadratic Discriminant Analysis (QDA), Linear Support Vector Machine (Linear SVM), radial SVM and Random Forest Classifier (RFC). (a) Mean accuracy for the same splits for each method. Random Forest (RFC) shows the best performance, followed by LDA. (b) Cross-validation accuracy averaged on the 5 splits. (c) Accuracy of the final models on multimodal nonlinear microscopy measurements of head and neck tissue (prediction dataset 1). LDA keeps a good performance on the prediction dataset, while RFC shows overfitting.

In addition, we have underlined that the main goal of the method is not to perform a binary classification, but to build quality rankings for the selection of the best correction. Indeed, an excellent classification would probably require a bigger training dataset and a more complex model. However, an excellent quality ranking can be reached also with light model like LDA: regardless of the class label, the quality score can be used to determine which image is the best (i.e., which image moves in the good direction in the feature space).

We have underlined this aspect in the manuscript as follows:

The trained model will be used not only to detect the presence of uneven illumination in raw images and determining the successful removal of the artifacts after processing, but more importantly to assign a score to every correction and automatically rank the images.

And:

LDA is a simple and efficient approach known for low risk of overfitting, ideal to obtain a generalizable model when the training dataset is limited. Moreover, LDA classifies the images by generating an interpretable decision score, which is the linear combination of the input features weighted by class coefficients and means (Supplementary Fig. 2). The score can be used for the generation of quality rankings (see the Methods section).

3. **Model Generalizability:** The LDA model was trained on a limited dataset comprising 40 single-channel images from a single tissue type (head-and-neck) and evaluated on just 23 three-channel images from the same tissue type and lab. This raises concerns about the model's generalizability. How consistent are the model's predictions when applied to different tissue types or images from other labs?

We value the Reviewer's concern about the dataset size. We have selected the LDA approach thanks to its good generalizability when trained with small datasets. We have included this in the main manuscript as specified in the previous point.

Regarding the prediction dataset, we would like to improve the transparency of our approach by adding further description to underline that EVEN is tested on prediction datasets that are completely independent from the training dataset. In particular, the datasets utilized in our study are:

- a. Training dataset: Multimodal images of biological specimens, including human nodular basal cell skin carcinoma, including CARS, TPEF and SGH measurements (2017, ref 9)
- b. Prediction dataset 1: multimodal measurements of human head and neck tissue slices, including CARS, TPEF, SHG measurements (2024, ref 18)
- c. Prediction dataset 2: multichannel measurements of stained cells by fluorescence microscopy (2024)
- d. Prediction dataset 3, 4: multichannel measurements of tissue culture cells by fluorescence microscopy and brain tissue slices (2017, ref 10)
- e. Prediction dataset 5 (added during the review process): widefield timelapse movie of differentiating mouse hematopoietic stem cells (2016, ref 10).

To highlight this variability of the datasets we have expanded the Introduction section as follows:

The application of EVEN is particularly advantageous for multi-modal and multichannel images, where each channel might require an individual optimization. In this manuscript, we demonstrate EVEN's effectiveness and generalizability by applying it to images processed by established correction methods and including datasets from varied sources.

In the Discussion:

These computational tools can be used to incorporate other markers into the method to suit other imaging experiments and transfer the method to any artifact type. For example, the edge energy ratio can be balanced in the horizontal and vertical directions to model a specific artifact more effectively, as shown in Supplementary Fig. 34. In this way, the EVEN method can be tailored. However, the pre-trained EVEN method presented here already works across a variety of samples, image modalities and artifacts, despite being trained only on tissue sections measured with multi-contrast microscopy.

And the Methods:

Our study includes a variety of datasets acquired by different laboratories and with different techniques, including multimodal nonlinear imaging (CARS, TPEF, SHG) and fluorescence microscopy, measurements of tissues and cells.

We have included the new dataset introduced during the peer-review in the Methods:

*Prediction dataset 5: timelapse movie of differentiating mouse hematopoietic stem cells.*⁷ We applied EVEN to a timelapse movie of 100 frames imaging the differentiation of mouse hematopoietic stem cells. The images are characterized by lateral shading artifact on the right edge and temporal flashing due to changes in the microscopy settings during the measurement. We corrected the movie with BaSiC applying different hyperparameters: no temporal drift with automatic options, temporal drift with lambda flat parameter equal to 0, 0.5 (automatic), 2, 4, 6. Then we built a composite image by stitching the frames in a 10x10 grid and we ranked the corrections using EVEN. Due to the lateral shading, we calculated a different energy ratio for this dataset, as explained below.

4. Evaluation Metrics: In addition to the LDA score, two metrics were proposed: intensity-based E_edge and spectrum-based P+. E_edge assumes homogeneous intensity distribution, which might not hold true in many practical cases. For example, when few tiles are present, edge tiles significantly influence the metric, reducing its validity. Intensity variations within tissue foreground (e.g., vessels) could further affect the metric. Why not use overlap differences from stitched regions, as done in the original CIDRE and BaSiC papers?

We thank the Reviewer for underlining an aspect of quality definition that we have not explored in the manuscript. The Reviewer is right: the methods associated with uneven illumination correction show optimal performance with a sufficient number of tiles. However, this is a problem faced during the correction of the images rather than the application of EVEN, which does not represent a bottleneck in the data pipeline.

Specifically, EVEN requires a composite image with at least 2 tiles per side for the computation of the peak prominence, while the edge energy can be computed also for a single tile. If these parameters are used to compare images with different objects, such statistics is not sufficient for a fair evaluation. However, the main application of EVEN is the comparison of different versions of the same image. Therefore, even with a low number of tiles, EVEN can identify a trend between the corrections and build a ranking.

We have added a simple example (Supplementary Fig. 34) to demonstrate the good performance of EVEN in case of a 2x2 crop of the image of Fig. 4:

Supplementary Fig. 4 - Evaluation and Enhancement of a multimodal image of human head and neck tissue slice on a small 2x2 tile region. One of the multimodal images of prediction dataset 1 is automatically optimized by EVEN. (a) The raw input measurement with strong uneven illumination. The small image region is extracted from the region marked by the white frame in the full measurement. (b) The three channels are corrected independently with three different correction methods: BaSiC, CIDRE, Fourier. Due to the different intensity distribution in the channels, that are generated by the different morphological features of the different components of the tissue, the correction methods show a different performance for different channels. Indeed, the multimodal corrections obtained by single methods (bottom row) are different and show residual artifacts. (c) The optimized correction is predicted by EVEN after the computation of the quality metrics for each correction of the channels. The evaluation differs from the result of Fig. 4 due to the different properties of this local region of interest. Scale bar: 200 μm .

To describe this aspect, we have modified our discussion as follows:

[...] EVEN allows comparison of different corrections by using images with a few tiles per side and does not limit the evaluation for practical cases with limited number of tiles.

And specified the requirements of the quality metrics in the Methods:

EVEN has basic requirements for the images to be processed: it should be applied to stitched images with a minimal number of tiles of equal size for the correct computation of the metrics. The evaluation can be executed before the final stitching if the generation of the composite image affects the regular periodicity. EVEN's performance might be impacted by shading artifacts with limited periodicity, such as irregular stripes and gradually varying shading. [...]

The edge energy can be computed on any number of tiles. [...]

The peak prominence can be computed on a small composite image but requires few tiles for the generation of a basic grid (see Supplementary Fig. 34).

Regarding the possibility of using the overlap of stitched regions, our aim is to provide a straightforward and repeatable approach to quality assessment. However, the use of overlapped regions would only be applicable to measurements with large overlaps. It would also require significant manual tuning to locate all overlaps in a large composite image. Therefore, although this method is

reliable and effective, it does not provide fully automated evaluation in real experimental scenarios and is often time consuming. In addition, the use of overlap regions might be particularly challenging for asymmetric vignetting, which is typical of point-scanning systems and nonlinear images. In this case, the evaluation of a small overlap region of the FOV might miss a comprehensive quality assessment.

We understand the importance of this approach and we have included more details in the introduction:

This issue is underlined by previous studies, where the comparison between different correction methods was assessed visually or by using overlapping images.^{1,8} **Visual assessment from an expert evaluator can be valid, but not objective and quantitative, while the latter approach is time consuming and requires significant overlap between tiles; therefore, it is a very effective approach for validation of new methods, but it is not practical in real-case experimental scenarios due to the more complex implementation and the need of evaluating strong asymmetries in the FOV that extend further than the overlap region, especially for point-scanning systems and nonlinear techniques.**

5. The metric P+ assesses periodic grid artifacts and can only evaluate stitched images. However, it might be inherently biased towards Fourier-based methods, as both the metric and Fourier method operate directly in the frequency domain.

We thank the Reviewer for the in-depth observation about our method, and we understand the Reviewer's concern about the connection between positive prominence and Fourier method.

We try to mitigate any bias by using one metric operating in the direct domain (edge energy) and one metric operating in the frequency domain (P+).

The Fourier method does not operate directly on the power spectrum, but it applies a prior logarithmic transformation to the image before the computation in the Fourier domain for optimization purposes. Thus, there is not exact correspondence between the Fourier method and our computation of the peaks.

Nevertheless, we are aware of this aspect, and we monitored the results without finding any unreasonable bias. Supplementary Fig. 4 shows that Fourier method obtains the best scores for the CARS channel but is largely surpassed by BaSiC and CIDRE for the SHG and TPEF channels. In addition, Supplementary Fig. 5 shows that the prominence of Fourier method has a variable trend, and the high score of the CARS channel is probably due to a high edge energy.

The public repository includes also further definitions of prominence, like negative peak prominence and magnitude prominence (sum of positive and negative prominence), allowing further experimentation by the user. We have now included a reference to these options in the methods.

6. Assessing correction performance remains challenging. Showing improvements in downstream tasks, such as cell segmentation or biologically relevant quantification, could strengthen the paper significantly.

We thank the Reviewer for underlining this important aspect and we acknowledge the importance of demonstrating a successful improvement of image analysis after the application of EVEN.

To address this issue, we have investigated cell segmentation for the three-channel measurement of stained cells (Fig. 5). We have applied the Cellpose software to the raw image, the single-method

corrections and the EVEN optimization. The result is now included in Fig. 5, which has been improved to include a visualization of the EVEN prediction and the downstream analysis.

Cellpose allows the prediction of cells outlines with a pretrained model and requires at least one channel imaging the cells cytoplasm and – optionally – the nuclei. Our measurements contain in the three channels different components of the cytoplasm. Therefore, we have given as input to Cellpose the sum of the three channels for each version of the corrected image (see the new Fig. 5 (b) for the complete workflow). This allows us to generate results also for the complete EVEN prediction and to compare them with the other options.

From the new Fig. 5 (d) we observe that EVEN not only provides a better visualization of the region of interest, together with enhanced colour balance, but it also generates an improved cell segmentation: especially in the regions highlighted by the dashed white frames, Cellpose identifies a higher number of cells in the EVEN optimization, with better performance compared to other corrections. This segmentation shows that the difference between raw and corrected images is obvious. In addition, although the difference between the correction methods is not always remarkable, an improvement in the segmentation result can be identified in specific regions of the EVEN optimization.

The manuscript has been adapted by updating Fig. 5:

Fig. 4 - Evaluation and Enhancement of experimental measurements of stained cells. A three-channel measurement of stained HEK293 cells measured by Ph2 objective is automatically optimized by EVEN (prediction dataset 2, red: peroxisomal

proteins (GFP); green: TOMM20 protein; blue: peroxisomal proteins (anti-GFP nanobody)). (a) Raw multi-channel image. The inset shows the 2x2 tile section of the image used in this figure for further visualization of the results. Multiple corrections are obtained by applying BaSiC, CIDRE, Fourier methods, and then optimizing the multi-channel image with EVEN. EVEN selects CIDRE for the red and green channel, and Fourier for the blue channel. (b) Steps to analyse the measurements of stained cells: the multi-channel images are converted to greyscale by summing the single channels (that contain signals from different components of the cytoplasm) and are analysed with automatic cells segmentation using Cellpose⁹. The greyscale image is obtained for the raw measurement, the single-channel corrections and the EVEN optimization. (c) Intensity sum (along y) of the greyscale inset for each method. The black dashed line indicates the border between neighbouring tiles. The corrected images show higher intensities at the edges of the tiles and the enhancement of sample features. EVEN and CIDRE show the greatest intensity recovery between tiles. (c) Top row: multi-channel images obtained with single-method corrections and EVEN optimization; the white dashed boxes highlight two regions significantly improved by EVEN. Bottom row: Cellpose prediction on the greyscale sum of the three channels for each method. After correction of uneven illumination, Cellpose can outline a greater number of cells, especially at the borders of neighbouring tiles. White dashed boxes highlights three regions where EVEN optimization provides better identification of the cells compared to non-optimized images. Scale bar: 180 μm , size of a single tile.

In addition, we have updated the Results accordingly and we have included the description of the analysis in the Methods:

We applied automatic cell segmentation with Cellpose⁹ to the fluorescence measurements of stained HEK293 cells acquired by Ph2 objective with the pretrained Cyto3 model¹⁰. Cyto3 predicts cell segmentations on measurements of the cytoplasm. Since the measured images contain signal from different components located in the cytoplasm, we applied Cellpose on the greyscale sum of the three channels with input cell diameter of 20 px. The sum provides better segmentation than single-channel thanks to complementary signal from different components and allows a comparison between the EVEN optimization and the single-method corrections.

We have also computed a comparison between the cell counts generated by Cellpose before and after the corrections, included now in Supplementary Fig. 31. The difference between good corrections is limited, but there is a significant improvement in the object count after the artifact removal.

(a) Stained cells – Ph2 objective (Fig. 5)

(b) Stained cells – Oil objective (Supplementary Fig. S28)

Supplementary Fig. 5 - EVEN score and Cellpose segmentation for multi-channel measurements of stained cells acquired with Ph2 objective (a) and Oil objective (b). Left column: normalized EVEN score computed for the EVEN optimization, single-method corrections and raw images. The reported score is the sum of the maximum normalized score of the three channels. Central column: number of cells counted by Cellpose for full-size multi-channel corrections. Right column: normalized cell count predicted separately for each tile by Cellpose. The removal of uneven illumination enables the identification of a higher number of cells.

7. **Parameter Tuning:** The authors propose using EVEN for tuning correction parameters, such as in BaSiC. Although promising, the study fixes the smoothness regularization parameters for flatfield and darkfield (λ -flat and λ -dark) at 0.5 without exploring other values. It would be beneficial to test different parameter settings, including turning off the darkfield correction and using flatfield correction alone.

We understand the Reviewer's request of additional applications of EVEN to hyperparameter tuning. We acknowledge the importance of an example of this practical application and include a further optimization in the Supplementary Fig. 34. We have utilized the time-lapse movie provided by the authors of BaSiC to compare different correction options of BaSiC. This dataset shows a temporal drift in the illumination, as well as a lateral shading, and we compare the use of the temporal drift correction with different λ parameters, as well as the correction without temporal drift correction. We generated a quality ranking with EVEN to select the optimal option.

The result is the following:

Supplementary Fig. 34 – EVEN optimization of BaSiC hyperparameters for a timelapse movie with temporal drift of the intensity. (a) The differentiation of mouse hematopoietic stem cells is measured in a timelapse movie with 100 frames. The frames show shading on the right side and a temporal change of the intensity. Therefore, they require an optimized correction with BaSiC, exploiting the Temporal Drift Correction (TDC) option and tuning of the λ parameter. We applied BaSiC without TDC ($\lambda=0.5$), and with TDC ($\lambda=0, 0.5, 2, 4, 6$), then we generated a composite image by stitching the frames and we applied EVEN evaluation. Due to the lateral shading, we selected a vertical threshold to identify the shaded area for the edge energy. (b) Quality ranking computed by EVEN. For a clear visualization of the residual artifacts, we show the cropped region highlighted by a yellow frame in the bottom-right panel. EVEN places at the top of the ranking corrections with no or minimal residual artifact, then the correction without TDC, where uneven illumination is removed successfully, and finally the corrections with strong residual shading.

And we have added the new dataset in the Methods:

*Prediction dataset 5: timelapse movie of differentiating mouse hematopoietic stem cells.*⁷ We applied EVEN to a timelapse movie of 100 frames imaging the differentiation of mouse hematopoietic stem cells. The images are characterized by lateral shading artifact on the right edge and temporal flashing due to changes in the microscopy settings during the measurement. We corrected the movie with BaSiC applying different hyperparameters: no temporal drift with automatic options, temporal drift with lambda flat parameter equal to 0, 0.5 (automatic), 2, 4, 6. Then we built a composite image by stitching the frames in a 10x10 grid and we ranked the corrections using EVEN. Due to the lateral shading, we calculated a different energy ratio for this dataset, as explained below.

8. Processing Time: The processing time of EVEN compared to the original three correction methods should be reported to assess practical utility.

We thank the Reviewer for mentioning this essential aspect of every processing method. The processing time required by EVEN is primarily determined by the computation of the metrics and is strongly influenced by the size of the composite image. That said, EVEN's processing time does not represent the bottleneck in the data pipeline, as the correction methods usually require longer time to be executed.

For a transparent and repeatable execution of EVEN, we have updated the main workflow in our repository, by including the visualization of the computation time for every step of the workflow and for the whole notebook.

In addition, we have included the processing time of the image of Fig. 5 in Supplementary Table 1 and included a new section in the Methods:

Processing time

The execution time of all steps is estimated for the multimodal image of Fig. 4 and is reported in Supplementary Table 1, including the average time of 5 processing attempts. Computations were performed on a Windows 10 Pro workstation equipped with an AMD Ryzen Threadripper 3960X 24-Core Processor, 128 GB RAM, and an NVIDIA GeForce RTX 3090. The image size is 8400 x 13200 px before stitching and 7560 x 11800 px after stitching. Even evaluation include the reading time of the images and computation of the quality metrics, EVEN optimization is the time required to load the selected channels and merge them into a multimodal image.

	Processingtime / s						
Number of images	BaSiC	CIDRE	Fourier	EVEN: Evaluate	EVEN: Train	EVEN: Predict + optimize	EVEN: Notebook
Single channel	13.61	12.73	0.79 (1.75)	5.53 (1.97)	0.09	0.004	-
3 channels	40.83	38.19	2.37 (5.25)	15.74	0.09	3.89	41.47

Supplementary Table 1 – Processing time of correction methods and EVEN for a multimodal image of head and neck tissue of Fig. 4. The column contain, from left to right, the processing time required for different steps of the workflow: BaSiC (Fiji plugin), CIDRE (Fiji plugin), Fourier (function implemented in the public repository, with the internal time to compute the function and, in brackets, the time get the image from the GPU and apply final normalization), Evaluate (computation of the quality metrics), Train (training of the LDA model), Predict + optimize (Prediction of the image score and generation of the optimized multimodal image), Notebook (full execution of the workflow notebook of the open source repository). The single-channel row includes the time to process a single-channel image. The EVEN evaluation contains the time to evaluate the raw image and the corrections and, in brackets, the evaluation time for a single image. The prediction time is only 4 ms because no multi-modal optimization must be executed. The 3-channel row contains the correction time for three channels and the EVEN workflow executed on the 12 resulting single-channel images using the notebook in the public repository (https://git.photonicdata.science/elena.corbetta/even/-/blob/main/workflow_even.ipynb).

To clarify our findings, we have modified the Discussion as follows:

One drawback of our method is currently **the need of more than one available correction to generate optimized image**, which cannot be determined a priori from the raw measurement: [...]

However, when the optimal method cannot be determined beforehand, EVEN does not significantly increase computational time and can speed up the selection of the best result through automatic

optimization. EVEN is a valuable choice for enhancing image analysis and the final interpretation of the measured sample if real-time corrections are not required.

9. Minor Issues: The pie charts in Fig. 3f and 5h are derived from different data types (FACS vs. time-lapse analysis) but presented similarly. Using different presentations to clearly differentiate these underlying data types is suggested. Labelling inconsistencies (doc/docetaxel, control/ctrl, parental in Fig. 3b, and concentration of Docetaxel in Fig. 5a) should be addressed for consistency and clarity.

This remark does not seem to be related to the content of our manuscript. However, we are open to receive any further comment that the Reviewer might have missed.

Reviewer #2 (Remarks on code availability):

I only took a very quick look at the code, which seems ok to me. In the python notebook, it uses BaSiCPy but in the paper it is stated use BaSiC Fiji version. Maybe authors can add a comment to clarify this confusion.

We thank the Reviewer for taking the time to check our code. The BaSiC corrections used in the paper are obtained with the Fiji plugin due to the workflow established at the beginning of our study, with the parallel utilization of BaSiC and CIDRE plugin on Fiji. During the development of the python-based software tool, we have decided to include additional tools in our repository and reference also BaSiCPy implementation. In this sense, the code is not meant only for result reproducibility, but also to provide tools for uneven illumination correction.

To avoid any confusion, we have explained the reason of our choice in the readme file of the repository and in the notebook for image correction as follows:

The BaSiC corrections of the manuscript are obtained with the Fiji plugin, but we include an optional code for executing BaSiCPy to provide users with additional tools and increase visibility of the Python-based version.

Reviewer #3 (Remarks to the Author):

In the manuscript by Corbetta et al. entitled “Making uneven illumination EVEN: automatic Evaluation and Enhancement of vignetting corrections in multimodal images”, authors propose a machine-learning based method, named EVEN, to assess and optimize corrections of uneven illumination in optical microscopy. The uneven illumination is a very important issue in optical microscopy, and some results in this study are promising. However, key method details are missed and some important concerns need to be addressed.

Major concerns:

1. Since authors use established methods (BaSiC, CIDRE, and Fourier) for image correction, then the key novelty of this study relies on the two markers, the edge energy ratio and the positive prominence, and the general framework of multi-modal image processing. Therefore, I think the novelty of this study is limited.

We understand the Reviewer’s concern regarding the novelty of the study, and we acknowledge the importance of explaining clearly which are the innovative contributions of our manuscript.

As the Reviewer mentioned, our study makes propose a new framework for image quality assessment and image quality optimization, which is particularly advantageous for multimodal and multichannel measurements.

To the best of our knowledge, there are no automatic workflows for image quality assessment that are specifically designed for uneven illumination and require minimal intervention by the user. Indeed, due to the lack of ground truth in experimental scenarios, most of the studies still rely on visual assessment or exploit the comparison of overlapped regions between tiles. The former approach can be valid in case of an expert evaluation, but is not objective and quantitative, while the latter does not have a straightforward implementation, is time consuming, and requires large, overlapped regions.

Therefore, our study generates a robust workflow for automatic image quality assessment and quality optimization with a machine learning-based combination of quality markers. EVEN can be trained with small datasets (roughly a few dozen bad and good images) and can be generalised over many samples (tissue from different locations, stained cells) and image modalities (fluorescence microscopy, nonlinear microscopy, timelapse measurements). The workflow can then be utilized with the pretrained model to generate automatic prediction of image quality, as discussed in the manuscript, or easily adapted with a set of markers defined by the user.

EVEN can be applied at any stage of the image data pipeline to ensure fair data management and correct data analysis. To underline these aspects and the contribution of our study, we have modified the introduction as follows:

To the best of our knowledge, there are nowadays no established, **automated** pipelines for the quantification of uneven illumination and for an objective evaluation of raw and processed images. [...] **In this paper, we present a machine learning (ML)-based automated workflow for quantitatively assessing uneven illumination in raw and corrected images.** The method, named EVEN, is intended to Evaluate the images of interest, and use the result to Enhance their quality by **determining which is the optimal result among the corrections generated**. Therefore, EVEN is **a quality evaluation and optimization tool** that makes the intensity distribution uniform not only from a visual point of view, but also based on a quantitative and objective assessment.

And we included main advantages of EVEN in the Discussion:

EVEN requires minimal effort by the user, as it relies only on basic knowledge about the tile size for full automation. If the uneven illumination pattern is more complex than usual vignetting, the user can tune the quality markers and opt for a refined approach.¹¹ In addition, EVEN allows comparison of different corrections by using images with a few tiles per side and does not limit the evaluation for practical cases with limited number of tiles. To facilitate the implementation of our method and encourage the integration of EVEN in the experimental workflow, we provide all the essential tools in a public repository.

2. It is not clear why these three established methods are selected. Authors do not mention the rationale of choosing these methods, especially when there are lots of emerging methods in dealing with uneven illumination.

We thank the Reviewer for the comment, and we aim to provide full transparency for our choice. We selected three established methods that are easily implementable (e.g, they provide open-source plugins) and are therefore likely to be selected by the community when dealing with uneven illumination problem. In addition, we have selected the methods by including different approaches from the direct and Fourier domain. As stated in the manuscript, the goal is not to determine the absolute performance of these methods, but rather to use these methods to demonstrate the applicability of EVEN in real processing cases. While we excluded simpler implementations like gaussian smoothing or tile average for the estimation of the mask, we also excluded emerging methods.

To make our choice clearer, we have modified the Results as follows:

The raw images are corrected by three established methods in the direct and Fourier domain (BaSiC⁷, CIDRE⁸ and Fourier method^{1,12})

And the Methods section:

The selected methods represent established and easily applicable tools, working in the direct and Fourier domain. This choice is only intended to validate EVEN's effectiveness and not to provide all the processing options, whose choice is much larger and in continuous development.

3. Authors mention that they built quality rankings from the model output thanks to a simple interpretation of the decision score. However, it is not clear how they define and obtain the decision score, and how this score balances the two main effects of uneven illumination on measured images, i.e., vignetting and the mosaicking effect.

We thank the Reviewer for the comment, as we consider essential the transparency of our method. EVEN is based on Linear Discriminant Analysis classifier. The decision function of the LDA model describes the decision rules of the classifiers and are defined as linear combinations of the input features (in our case, the quality metrics). Each class is associated to a decision function, whose weights and offsets are adjusted during the training process. During the prediction process, the features associated with the input image are inserted in the decision function and the decision score of the image is obtained.

In line with the Reviewer's request, we have expanded the Methods section including a detailed description of the decision score:

Step 2.2: interpretation of the model output and generation of quality rankings. An in-depth automatic evaluation workflow can be generated by interpreting the model output. The classification rules of the LDA classifier are described by the decision functions $\delta_k(x)$ defined for every class k and computed for the input sample x to be classified. The decision functions are defined as linear combinations of the input features x_f (i.e., the quality metrics computed for image x), with coefficients $c_{k,f}$, means $\mu_{k,f}$, and class priors π_k that are determined by the training process:

$$\delta_k(x) = \sum_{x_f} \left(x_f c_{k,f} - \frac{1}{2} \mu_{k,f} c_{k,f} \right) + \log \pi_k$$

The decision scores for the image to be predicted is calculated by inserting the quality metrics in the formula above. In case of a binary classification, the images are assigned to the good or bad class according to a single decision score returned by the LDA model that represents the difference $\delta_{good}(x) - \delta_{bad}(x)$, and it results to a positive value for images assigned to the good class and a negative value otherwise. The weighting coefficients and means to define each class's decision function represent the relative importance given to each quality metric. A positive value of the coefficients is assigned to features whose value increases for high quality images, whereas a negative value is associated to features increased by the presence of uneven illumination, but they might have a complex behaviour to balance the effect of multiple features. In the simple case of two quality metrics, as expected, the edge energy ratio is weighted with a positive coefficient, meaning that an increase of this metric is related to a probability increase for the good class, and the opposite applies to positive prominence. In addition, the mean of the bad class is characterized by low edge energy and high positive prominence, while the mean of the good class shows opposite behaviour (Supplementary Fig. 2).

4. An important question that needs to be answered is that how will the correction affect subsequent analysis. As authors mentioned in the abstract, the uneven illumination would impede the image analysis. I believe that demonstration of how the image analysis would be improved following correction of uneven illumination, by additional experiment or analysis, would greatly strengthen this study.

We thank the Reviewer for highlighting the need of further validation of our method with subsequent analysis. We agree with the Reviewer that further image analysis can significantly strengthen the conclusions about EVEN.

We have investigated cell segmentation for the three-channel measurement of stained cells (Fig. 5) by applying the Cellpose software to the raw image, the single-method corrections and the EVEN optimization, including the results in the new version of the image. Cellpose predicts cells outlines by using at least one channel measuring the cells cytoplasm and –optionally–the nuclei. Our multicolour measurements contain different components of the cytoplasm in each channel. Therefore, following the workflow depicted in Fig. 5 (b), we have given as input to Cellpose the sum of the three channels for each version of the corrected image. This allows us to generate results also for raw image, single-channel corrections and EVEN prediction.

From the new Fig. 5 (d) we observe that EVEN not only provides a better visualization of the region of interest, together with enhanced colour balance, but it also generates an improved cell segmentation: especially in the regions highlighted by the dashed white frames, Cellpose identifies a higher number of cells in the EVEN optimization, with better performance compared to other corrections. This segmentation shows that the difference between raw and corrected images is obvious. In addition,

although the difference between the correction methods is not always remarkable, an improvement in the segmentation result can be identified in specific regions of the EVEN optimization.

Fig. 5 - Evaluation and Enhancement of experimental measurements of stained cells. A three-channel measurement of stained HEK293 cells measured by Ph2 objective is automatically optimized by EVEN (prediction dataset 2, red: peroxisomal proteins (GFP); green: TOMM20 protein; blue: peroxisomal proteins (anti-GFP nanobody)). (a) Raw multi-channel image. The inset shows the 2x2 tile section of the image used in this figure for further visualization of the results. Multiple corrections are obtained by applying BaSiC, CIDRE, Fourier methods, and then optimizing the multi-channel image with EVEN. EVEN selects CIDRE for the red and green channel, and Fourier for the blue channel. (b) Steps to analyse the measurements of stained cells: the multi-channel images are converted to greyscale by summing the single channels (that contain signals from different components of the cytoplasm) and are analysed with automatic cells segmentation using Cellpose⁹. The greyscale image is obtained for the raw measurement, the single-channel corrections and the EVEN optimization. (c) Intensity sum (along y) of the greyscale inset for each method. The black dashed line indicates the border between neighbouring tiles. The corrected images show higher intensities at the edges of the tiles and the enhancement of sample features. EVEN and CIDRE show the greatest intensity recovery between tiles. (c) Top row: multi-channel images obtained with single-method corrections and EVEN optimization; the white dashed boxes highlight two regions significantly improved by EVEN. Bottom row: Cellpose prediction on the greyscale sum of the three channels for each method. After correction of uneven illumination, Cellpose can outline a greater number of cells, especially at the borders of neighbouring tiles. White dashed boxes highlights three regions where EVEN optimization provides better identification of the cells compared to non-optimized images. Scale bar: 180 μm , size of a single tile.

Moreover, we have included in the supplementary material the overall Cellpose result, obtained on the measurements of Prediction dataset 2 with the two objectives:

Supplementary Fig. 6 - EVEN score and Cellpose segmentation for multi-channel measurements of stained cells acquired with Ph2 objective (a) and Oil objective (b). Left column: normalized EVEN score computed for the EVEN optimization, single-method corrections and raw images. The reported score is the sum of the maximum normalized score of the three channels. Central column: number of cells counted by Cellpose for full-size multi-channel corrections. Right column: normalized cells count predicted separately for each tile by Cellpose. The removal of uneven illumination enables the identification of a higher number of cells.

Specific comments:

- The title could be further improved. I wonder why authors mention “vignetting” only in the title. We would like to remove any confusion related to the terminology used in our manuscript. The term vignetting has been utilized because it refers to the artifact affecting the single tiles if the composite images (and it is therefore responsible for the generation of the mosaic). However, we understand that this term might be too specific, and we have changed the title of our manuscript using the more general and common term *flat-field correction*.
- I do not think that putting Fig. 1 in Introduction is proper. The authors tend to mention a general pipeline, while Fig. 1 is specific to EVEN (the method proposed in this study). Rather, I think it fits to Results better. We understand the Reviewer’s concern and to make our message more transparent we have modified Fig. 1 (b) by substituting the term ‘EVEN’ with ‘Image quality assessment’. Indeed, every image quality assessment method for uneven illumination can be used for the tasks highlighted in the Fig. 1. Our method is now outside the main box, meaning that it can be used as image evaluation tool to execute the mentioned tasks:

Fig. 6 - Experimental pipeline for optical imaging of large samples in presence of uneven illumination. (a) The experimental pipeline for optical microscopy is composed of image reconstruction, processing, and analysis steps to extract quantitative information from the measured sample and address the questions raised by the research study. When uneven illumination affects the measured images, the pipeline must be adapted to remove this artifact and enable an accurate image analysis. In the case of measurements of large samples, multiple tiles of the tissue are stitched into a composite image and flat-field correction is applied. Finally, the enhanced image can be analysed to retrieve the information of interest. (b) Image quality assessment should be always integrated in the experimental pipeline for a quantitative evaluation of the raw and corrected images. It can be utilized for automatic detection of uneven illumination, fine-tuning of the hyperparameters of different correction methods and the selection of the optimized correction before image analysis. EVEN can be used as a reliable and automatic tool for these tasks.

7. I do not quite understand Fig. 3c. How do you select “Good” and “Bad” data? Why do authors write “visual evaluation” in the figure? These concerns also apply for the analysis of the positive prominence metric (Fig. 3d-f). Authors are suggested to provide the context.

We thank the Reviewer, and we try to remove any confusion by expanding the description in our manuscript. The good and bad images are taken from a dataset provided by the study of Chernavskaia et. al, and have been split into the two groups by using the previous expert evaluation provided by the authors and with further visual selection by us for selecting a balanced subset among the available images. As the images come from a real experimental case and have been processed to ensure uneven illumination removal, the ground truth is not available and visual evaluation is necessary. We have expanded the first subsection of the Methods as follows:

The training dataset for binary classification is composed of 40 good and 40 bad experimental images from the previous study of Chernavskaia et al.¹, which investigated different correction methods in the direct and spatial frequency domain and provided an expert evaluation of all results. In our training dataset, the bad images are experimental stitched single-channels CARS or TPEF measurements of biological specimens affected by uneven illumination. The good images are good corrections among the results provided by the study, selected by visual assessment and according to the visual evaluation

labels provided by the previous authors. Some samples are included in both classes, but not all the images are paired. Supplementary Fig. 1 shows a subset of the training dataset, to demonstrate the high content and structural variability despite the limited number of images.

And we have modified the label 'Experimental dataset' to 'Known experimental dataset' in Fig. 3 (c) and Fig. 3 (f), updated the figure caption, and updated the Results as follows:

Fig. 1 (c) and Fig. 1 (f) show the statistical difference of the metrics computed for a known experimental dataset composed of 40 measurements affected by strong uneven illumination and 40 high-quality images that were well corrected by different processing methods (see subset of the images in Supplementary Fig. 1 and details about the correction in the Methods).

8. Is the positive prominence metric dependent on other properties of the image, such as the size? Also, do authors sum all the prominences to acquire this metric?

We thank the Reviewer for this in-depth comment on the metrics. The positive prominence $P+$ is defined as the sum of the periodic peaks identified in the sum-normalized power spectrum. Therefore, some considerations can be provided:

- $P+$ increases with the number of peaks. From the definition of the Fourier transform algorithm in NumPy, the number of peaks is equal –in number– to the period of the mosaic. Therefore, for an internal comparison of a dataset acquired with same acquisition parameter this dependency can be ignored.
- $P+$ is computed for a sum-normalized power spectrum. Therefore, the total area under the power spectrum is normalized to 1 and the area under the peaks represents the fraction of image energy allocated to the mosaic periodicity. This aspect mitigates any variability generated in the number of peaks from the considerations in the previous point.

To clarify these aspects, we have expanded the Methods as follows:

Sum of the prominence of periodic peaks in the power spectrum. [...] For a composite image of size $N \times M$ pixels and composed by square tiles of $T \times T$ pixels, the mosaic has a period T , and T peaks are generated in the power spectrum located at frequencies $f_x = \pm \frac{k_x N}{T}$, $f_y = \pm \frac{k_y M}{T}$ for $k = 1, \dots, \frac{T}{2}$. [...]

The sum normalization ensures that the peak prominence represents the fraction of image energy located at the frequencies related to the mosaic and decreases the metric variability across different image sizes. The peak prominence can be computed on a small composite image but requires few tiles for the generation of a basic grid (see Supplementary Fig. 35).

9. I highly recommend that authors re-organize some contents. For example, in Fig. 3c and Fig. 3f, they show the statistical difference of the metrics computed for 40 experimental images affected by strong uneven illumination and 40 experimental images that were well corrected by processing methods; however, they have not yet detailed how they perform the image enhancement/correction until Fig. 4.

We understand the Reviewer's concern about providing more details about how our method is implemented. As mentioned in question 2 and question 7, we have included a more detailed description about our choice of the correction methods, updated the description of Fig. 3 and the description of the training dataset. The correction methods are used in our manuscript only as example

options before the application of EVEN, therefore they have been introduced later in the manuscript for specific applications.

We hope the current modifications provide sufficient clarity, but we are open to further modifications requested by the Reviewer.

10. It looks that the sample size for LDA training is quite limited, which might impact the accuracy of the model.

We understand the importance of this aspect raised by the Reviewers. Our training dataset can be considered limited but, as showed in the new Supplementary Fig. 1, it is composed by a large variety of structures and samples to provide better generalizability of the metrics. Indeed, since the quantity of features is low and LDA provides low risk of overfitting, we can successfully extract valuable trends of the quality metrics from the training dataset.

We have explained this aspect in the Results as follows:

[...] we train a Linear Discriminant Analysis (LDA) model on a set of experimental single-channel nonlinear measurements, consisting of 40 good and 40 bad images composed of multiple tiles (workflow of Fig. 2(b) and Supplementary Fig. 1). [...] LDA is a simple and efficient approach known for low risk of overfitting, ideal to obtain a generalizable model when the training dataset is limited. Moreover, LDA classifies the images by generating an interpretable decision score, which is the linear combination of the input features weighted by class coefficients and means (Supplementary Fig. 2). The score can be used for the generation of quality rankings (see the Methods section).

And in the Methods:

Manual assessment and model training. The single-channel images used for the manual assessment of uneven illumination, the generation of semisynthetic images and the model training are part of an experimental dataset introduced by the previous work of Chernavskaia et al.¹, composed of CARS, TPEF and SHG measurements of different biological specimens, including human head and neck tissue sections², human skin³, human tissue biopsies from colonoscopy or surgical resection⁴, mice colorectal biopsies⁵, and pig brain tissue⁶ (Supplementary Fig. 1).

[...]

The good images are good corrections among the results provided by the study, selected by visual assessment and according to the visual evaluation labels provided by the previous authors. Some samples are included in both classes, but not all the images are paired. Supplementary Fig. 1 shows a subset of the training dataset, to demonstrate the high content and structural variability despite the limited number of images.

Moreover, a subset of the training dataset is included in the Supplementary information:

Supplementary Fig. 7 – Example images from the training dataset. Subset of square crops from the training dataset, with tile size equal to 512 px, variable number of tiles and a wide variety of imaged structures. The first four images from the top right are paired in the two classes. (a) Example images from the bad class: experimental uneven illumination. (b) Example images from the good class: good corrections of the experimental measurements.

11. Page 7, Paragraph 3, it should be Fig. 2(c).

We thank the Reviewer for carefully checking the references, we have corrected the text.

12. In Fig. 4, authors mention that Fourier results in the optimized correction for CARS. However, it is revealed from the resultant image that a lot of details within the CARS image are missed following Fourier processing, which might be a reason of the reduction in mosaic effect. Please clarify this.

We thank the Reviewer for careful inspection of the results. Indeed, Fourier method shows a lower average intensity in the CARS channel, with bright region in the bottom right part. However, the details are less visible with the original contrast of the figure, but they are not missing, as demonstrated in Response Figure 4 after strong contrast adjustment (same for all crops, extracted from a darker region in the Fourier CARS channel) just to enhance the visualization:

Response Figure 4 – Corrections of the CARS channel of the multimodal image from Fig.4. The crops are showed after equal contrast adjustment to improve the visualization of darker regions. Despite the darker average intensity of the Fourier correction, it is clear that uneven illumination is removed successfully without loss of details.

We understand that the Fig. 4 requires further explanation for an audience that cannot further inspect the result, and we have expanded the caption as follows:

[...] the Fourier method is selected as the best correction for the CARS signal, **as it provides flatter intensity distribution and enhancement of few bright regions without loss of details in the darker areas**, whereas CIDRE is selected for TPEF and SHG channels.

13. In Page 8, I wonder why authors mention “visually equivalent”, since they have quantitative metrics to evaluate the corrected images.

We thank the Reviewer for underlining this potential improvement in the result description. We have modified the text by adding some references to data already provided in the supplementary material: In few cases (Supplementary Fig. 8, 12, 22, 27) all the corrections are visually equivalent **and indeed most of their channels show a low variability in the decision scores of Supplementary Fig. 4 and quality metrics of Supplementary Fig. 5, demonstrating that EVEN do not generate unexpected results for high-quality images.**

14. It is good that authors show a huge number of examples in Supplementary Figures (Fig. S4-Fig. S26). However, for some raw images, I think they themselves are good enough (e.g., Fig. S25), and the optimization would not contribute to a better analysis of these images.

The Reviewer is right: some of the images do not show major degradation caused by uneven illumination. However, we decided to provide a full overview of the available images, and we think that it is beneficial to show the stable performance of the quality metrics and EVEN in presence of very low artifacts.

The modification mentioned in comment 13 provides a reference to the quantitative results for these images.

15. Authors mention that “Therefore, a fine-tuned optimization of the set of quality metrics might be required in case of unusual behaviour of the correction methods”; however, they did not propose how to do that and whether it will make sense. Authors are suggested to prove that.

We thank the Reviewer for underlining the need of a more detailed explanation of atypical scenarios. In line also with the requests of exploring hyperparameters tuning within a single correction method, we have included a further application of EVEN.

We have applied EVEN to the BaSiC correction of a time lapse movie of differentiating mouse hematopoietic stem cells which shows a different experimental artifact than those showed so far in

our manuscript. The images are affected by non-radial shading on the right edge and, additionally, by a temporal flashing caused by varying microscopy settings. Therefore, these images require careful correction exploiting the different options of the BaSiC software.

We have processed these images as explained in the new section added in the Methods:

*Prediction dataset 5: timelapse movie of differentiating mouse hematopoietic stem cells.*⁷ We applied EVEN to a timelapse movie of 100 frames imaging the differentiation of mouse hematopoietic stem cells. The images are characterized by lateral shading artifact on the right edge and temporal flashing due to changes in the microscopy settings during the measurement. We corrected the movie with BaSiC applying different hyperparameters: no temporal drift with automatic options, temporal drift with lambda flat parameter equal to 0, 0.5 (automatic), 2, 4, 6. Then we built a composite image by stitching the frames in a 10x10 grid and we ranked the corrections using EVEN. Due to the lateral shading, we calculated a different energy ratio for this dataset, as explained below.

Due to the lateral shading we have used the threshold as mentioned in the new Methods section:

In the case of non-radial vignetting, a different threshold can be set for E_{shaded} . For example, for a lateral shading on the left side of the tiles, we can define $E_{shaded} = \sum_{x_i > k_v} t_{sum}(x_i, y_i)$, by selecting all pixels with coordinate x larger than a vertical threshold k_v , which can be set equal to half the lateral tile size T . The obtained edge energy follows the same trend of the radial threshold and does not require further training with a new dataset.

In this manuscript, we have used the radial definition for all datasets, except for prediction dataset 5, which required the vertical threshold. The edge energy can be computed on any number of tiles.

And the result is reported in the supplementary material and referenced in the main text:

Supplementary Fig. 8 – EVEN optimization of BaSiC hyperparameters for a timelapse movie with temporal drift of the intensity. (a) The differentiation of mouse hematopoietic stem cells is measured in a timelapse movie with 100 frames. The frames show shading on the right side and a temporal change of the intensity. Therefore, they require an optimized correction with BaSiC, exploiting the Temporal Drift Correction (TDC) option and tuning of the λ parameter. We applied BaSiC without TDC ($\lambda=0.5$), and with TDC ($\lambda=0, 0.5, 2, 4, 6$), then we generated a composite image by stitching the frames and we applied EVEN evaluation. Due to the lateral shading, we selected a vertical threshold to identify the shaded area for the edge energy. (b) Quality ranking computed by EVEN. For a clear visualization of the residual artifacts, we show the cropped region highlighted by a yellow frame in the bottom-right panel. EVEN places at the top of the ranking corrections with no or minimal residual artifact, then the correction without TDC, where uneven illumination is removed successfully, and finally the corrections with strong residual shading.

16. In Fig. 5, to be honest, I could hardly tell which is better visually among EVEN prediction, BaSiC, and CIDRE, and more importantly, I do not believe that the subtle difference among these methods would really affect the subsequent analysis.

We thank the reviewer for underlining the need of an improved visualization. We have improved Fig. 5, that now includes also analysis results for cell segmentation.

We hope that the current visualization makes the interpretation of the figure straightforward, as we understand that the dark nature of the measurements makes the uneven illumination less visible. We could demonstrate that the use of EVEN can significantly improve the downstream analysis, allowing the identification of more cells and with more accurate outline (see Figure 5, included in the response to question 4).

17. In Fig. 5, authors should explain what the three channels stand for.

We thank the reviewer for the comment. We have specified the details of the three channels in the captions of all figures including prediction dataset 2, or directly in the figure labels. The three channels contain signal detected from the following structures: peroxisomal proteins (GFP) in the red channel, TOMM20 protein in the green channel, and peroxisomal proteins (anti-GFP nanobody) in the blue channel.

18. In Fig. 5b, authors mention that “The profiles clearly show that the periodic artifact in the raw images is strongly reduced by the correction methods”. However, although improved a little bit, the periodic artifact is still quite obvious as observed from these profiles, for all the three correction methods.

We thank the Reviewer for highlighting the need for better evidence of our findings. We have updated Fig. 5 with an improved and simplified visualization of the profiles (see the new figure included in question 4). We would like to underline that the quality of the correction is limited not by the performance of EVEN, but to the performance of the correction results available in this specific case. Despite the potential improvement in the correction, we demonstrated that the available corrections provide a remarkable improvement in the image analysis, and EVEN enhances the automatic cell segmentation thanks to the multi-channel optimization. This result shows that, in general, improving the correction quality leads to better image analysis.

19. In page 8, authors mention: “However, depending on the processing method, single channels might be enhanced differently, resulting in a variable tone in the multi-colour image. In this scenario, EVEN ensures the selection of channels with the top global quality features.” I appreciate this; however, I wonder what is the ground truth which can be used to evaluate the correction.

We thank the Reviewer for this comment and for acknowledging the importance of a balanced optimization of all channels in multi-colour measurements. The measurements used in our study do not have the associated ground truth for practical reasons. The ground truth could be generated by measuring the same samples with tiles of decreasing size to limit the vignetting effect; however, this would require longer acquisition time and repeated measurements on the same sample, which might degrade the tissue or cells. This problem has been recognized by previous studies and has been faced, for example, by evaluating the image quality by comparing overlapping tiles.

The channel balance is largely determined by the detectors, the measured signal and sample. Therefore, with the tools available to us we can assume that a good retrieval of flat-field illumination without residual shading and loss of artifacts is the best step to prevent any further image degradation. We have underlined this aspect in the discussion, and we have underlined that this is an important issue to be considered in further investigations:

Channel-based optimization balances the contribution of multiple colours in shaded regions, as uneven illumination varies with the detected signal due to sample, optics, and detector properties.

Reviewer #3 (Remarks on code availability):

I think that the the code is a usable resource for the community.

We thank the Reviewer for checking our code and acknowledging its usability for the community.

1. Chernavskaia O, Guo S, Meyer T, Vogler N, Akimov D, Heuke S, Heintzmann R, Bocklitz T, Popp J. Correction of mosaicking artifacts in multimodal images caused by uneven illumination. *Journal of Chemometrics* **31**, (2017).
2. Legesse FB, Chernavskaia O, Heuke S, Bocklitz T, Meyer T, Popp J, Heintzmann R. Seamless stitching of tile scan microscope images. *J Microsc* **258**, 223-232 (2015).
3. Heuke S, Vogler N, Meyer T, Akimov D, Kluschke F, Rowert-Huber HJ, Lademann J, Dietzek B, Popp J. Multimodal mapping of human skin. *Br J Dermatol* **169**, 794-803 (2013).
4. Chernavskaia O, Heuke S, Vieth M, Friedrich O, Schürmann S, Atreya R, Stallmach A, Neurath MF, Waldner M, Petersen I, Schmitt M, Bocklitz T, Popp J. Beyond endoscopic assessment in inflammatory bowel disease: real-time histology of disease activity by non-linear multimodal imaging. *Scientific Reports* **6**, 29239 (2016).
5. Bocklitz TW, Salah FS, Vogler N, Heuke S, Chernavskaia O, Schmidt C, Waldner MJ, Greten FR, Bräuer R, Schmitt M, Stallmach A, Petersen I, Popp J. Pseudo-HE images derived from CARS/TPEF/SHG multimodal imaging in combination with Raman-spectroscopy as a pathological screening tool. *BMC Cancer* **16**, 534 (2016).
6. Calvarese M, Corbetta E, Contreras J, Bae H, Lai C, Reichwald K, Meyer-Zedler T, Pertzborn D, Muhlig A, Hoffmann F, Messerschmidt B, Guntinas-Lichius O, Schmitt M, Bocklitz T, Popp J. Endomicroscopic AI-driven morphochemical imaging and fs-laser ablation for selective tumor identification and selective tissue removal. *Sci Adv* **10**, eado9721 (2024).
7. Peng T, Thorn K, Schroeder T, Wang L, Theis FJ, Marr C, Navab N. A BaSiC tool for background and shading correction of optical microscopy images. *Nat Commun* **8**, 14836 (2017).
8. Smith K, Li Y, Piccinini F, Csucs G, Balazs C, Bevilacqua A, Horvath P. CIDRE: an illumination-correction method for optical microscopy. *Nat Methods* **12**, 404-406 (2015).
9. Stringer C, Wang T, Michaelos M, Pachitariu M. Cellpose: a generalist algorithm for cellular segmentation. *Nature Methods* **18**, 100--106 (2021).
10. Stringer C, Pachitariu M. Cellpose3: one-click image restoration for improved cellular segmentation. *Nature Methods* **22**, 592-599 (2025).
11. Corbetta E, Bocklitz T. Machine Learning-Based Estimation of Experimental Artifacts and Image Quality in Fluorescence Microscopy. *Advanced Intelligent Systems*, (2024).
12. Bulan O, Buckley R, Wiegandt R, Sharma G. Correcting illumination variations in photomicrograph mosaics of daguerreotypes. *2012 IEEE International Conference on Acoustics, Speech and Signal Processing (ICASSP)*, 1685-1688 (2012).

We thank the Reviewers for their positive feedback on our manuscript revision and for acknowledging the usability of our code.

In the following, we provide our reply to the remarks of Reviewer #2. We include in **black** the comments of the Reviewer, in **blue** our response and in **red** the modifications applied to the manuscript. All the modifications are highlighted in **red** also in the manuscript and in the supporting file.

Additionally, we have modified the manuscript to address all the formatting issues raised by the Authors Checklist.

Reviewer #2 (Remarks to the Author):

The revision has addressed most of my previous comments. Only two relevant comments:

1. Classification accuracy vs ranking: I understand the authors rebuttal that the trained model is used for ranking, does not have to achieve 100% classification accuracy. Authors also add classification performance of other classifier (random forest, etc). How about check the consistency of different ranking using different classifier?

We thank the Reviewer for addressing this important aspect of our method. We have generated quality rankings for Prediction Dataset 1 to assess the consistency between Linear Discriminant Analysis and other classifiers. We have not computed rankings for Random Forest classifier because this method it is not suitable for the generation of quality rankings.

Over all channels of the 23 RGB images (that is, 69 predictions of the best correction), all classifiers generate an identical selection of the best image in 37 cases (54%). To evaluate the consistency of the quality rankings, we used Kendall's tau correlation coefficient. A quality ranking was generated for every image and channel, and Kendall's tau was computed between the LDA-derived ranking and those obtained from the other classifiers. The average correlation coefficients show good correlation between the rankings, with linear models scoring more than 75% overlap:

- LDA vs LDA: 1.00 (perfect correlation)
- LDA vs QDA: 0.72
- LDA vs Linear SVM: 0.76
- LDA vs Radial SVM: 0.68

This assessment shows higher agreement between linear classifiers, which generally show better performance across the dataset, confirming the findings already highlighted in the paper: linear methods should be preferred for simple classification tasks and limited training datasets to prevent overfitting.

We have expanded Supplementary Fig. 3 by including the correlation coefficients and two examples from samples that show relevant quality variability among different corrections:

Supplementary Fig. 1 – Comparison between machine learning (ML)-based classifiers. 5-fold cross validation is executed with Linear Discriminant Analysis (LDA), Quadratic Discriminant Analysis (QDA), Linear Support Vector Machine (Linear SVM), radial SVM and Random Forest Classifier (RFC). (a) Mean accuracy for the same splits for each method. Random Forest (RFC) shows the best performance, followed by LDA. (b) Cross-validation accuracy averaged on $n = 5$ splits. (c) Accuracy of the final models on multimodal nonlinear microscopy measurements of head and neck tissue (prediction dataset 1). LDA keeps a good performance on the prediction dataset, while RFC shows overfitting. *The consistency between quality rankings obtained by LDA and other classifiers is assessed through Kendall's rank correlation coefficient computed between the decision scores predicted by the models. The average correlation coefficients for $n = 69$ rankings are 0.72 for LDA vs QDA, 0.76 for LDA vs Linear SVM, and 0.68 for LDA vs Radial SVM. We report the decision scores of different classifiers for two samples of Prediction dataset 1: (c) sample 1, (d) sample 13 (Fig. 4 of main paper). The scores show good agreement between different classifiers. Mismatches are found mostly between images with similar quality and linear models show generally better performance and generalizability.*

And extended the main text as follows:

Compared to other ML-based classifiers, LDA shows good performance for both the 5-fold cross-validation with the training dataset and the prediction of unseen images, **as well as good agreement with alternative linear classifiers, which show good generalization for our problem** (Supplementary Fig. 3).

And in the Methods:

The consistency between the quality rankings of LDA and the alternative classifiers (Supplementary Fig. 3) is assessed using Kendall's rank correlation coefficient, calculated for each image and channel separately based on all decision scores, and then averaged over all rankings.

Additionally, we include the decision scores computed by all classifiers for Prediction dataset 1 as appendix to this document, for better readability. It is possible to observe that classifiers show similar trends in most cases, and linear classifiers show a consistent behaviour.

2. Reviewers add down-stream analysis like cellpose segmentation, which is nice and an important aspect. How about put some of quantification of cell segmentation (now Supplementary Fig. 5) into the main Fig. 4?

We thank the Reviewer for highlighting the importance of the Cellpose segmentation and we understand the quantitative supporting data are relevant to the main results section. We have modified Fig. 5 by including the EVEN score (reported also in Supplementary Fig. 31 (a)) and the cell count relative to the zoomed section showed in Fig. 5 (d).

While quantification is a fundamental aspect of image analysis, in the case of this dataset the ground truth is not available to compare predicted and true cell count. The true cell count would be extracted manually from the optimal corrected image and would be biased towards the EVEN result. In general, we expect correlation between the EVEN score and the cell count. However, the highest cell count does not necessarily indicate best image quality due to possible false positives.

After these considerations, we have decided to leave most of the quantitative analysis in the supplementary material, as it does not represent the strongest conclusion for our study. However, image and caption of Fig. 5 have been extended as follows:

Fig. 1 - Evaluation and Enhancement of experimental measurements of stained cells. A three-channel measurement of stained HEK293 cells measured by Ph2 objective is automatically optimized by EVEN (prediction dataset 2, red: peroxisomal proteins (GFP); green: TOMM20 protein; blue: peroxisomal proteins (anti-GFP nanobody)). (a) Raw multi-channel image. The inset shows the 2x2 tile section of the image used in this figure for further visualization of the results. Multiple corrections are obtained by applying BaSiC, CIDRE, Fourier methods, and then optimizing the multi-channel image with EVEN. EVEN selects CIDRE for the red and green channel, and Fourier for the blue channel. (b) Steps to analyse the measurements of stained cells: the multi-channel images are converted to greyscale by summing the single channels (that contain signals from different components of the cytoplasm) and are analysed with automatic cells segmentation using Cellpose¹⁹. The greyscale image is obtained for the raw measurement, the single-channel corrections and the EVEN optimization. (c) Intensity sum (along y) of the greyscale inset for each method. The black dashed line indicates the border between neighbouring tiles. The corrected images show higher intensities at the edges of the tiles and the enhancement of sample features. EVEN and CIDRE show the greatest intensity recovery between tiles. (d) Top row: multi-channel images obtained with single-method corrections and EVEN optimization; the white dashed boxes highlight two regions significantly improved by EVEN. Bottom row: Cellpose prediction on the greyscale sum of the three channels for each method. After correction of uneven illumination, Cellpose can outline a greater number of cells, especially at the borders of neighbouring tiles. White dashed boxes highlight three regions where EVEN optimization provides better identification of the cells compared to non-optimized images. Bottom labels show, for each image, the normalized EVEN score summed over three channels and the cell count in the zoomed region. While the count is not strictly correlated with segmentation performance, a good correction of uneven illumination enhances downstream analysis and generally increases the number of detected cells. Further quantification is provided in Supplementary Fig. 31. Scale bar: 180 μ m, size of a single tile.

We have added in the Methods section a small description on how the EVEN score is calculated for the images:

The normalized EVEN score is computed by z-score normalizing the decision score of single-channel images over the dataset and summing the decision score of single channels for every multi-channel image.

Appendix: decision scores predicted by different classifiers for every sample.

Response Figure 1 - Decision function computed by different classifiers for samples 1 to 6 of Prediction dataset 1.

Response Figure 2 - Decision function computed by different classifiers for samples 7 to 12 of Prediction dataset 1.

Response Figure 3 - Decision function computed by different classifiers for samples 13 to 18 of Prediction dataset 1.

Response Figure 4 - Decision function computed by different classifiers for samples 19 to 23 of Prediction dataset 1.